# Diagnostic Delay in Soft Tissue Sarcomas: A Review

**DOI:** 10.3390/cancers17111861

**Published:** 2025-05-31

**Authors:** Juan Ángel Fernández, Beatriz Gómez, Daniel Díaz-Gómez, Irene López, Pablo Lozano, Paula Muñoz, Francisco Cristóbal Muñoz-Casares, Vicente Olivares-Ripoll, Hugo Vasques, José Manuel Asencio-Pascual

**Affiliations:** 1Department of General Surgery, Hospital Universitario Los Arcos del Mar Menor, 30739 Murcia, Spain; beatriz_92gp@hotmail.com; 2Campus de los Jerónimos, Universidad Católica de Murcia UCAM, 30107 Murcia, Spain; 3Servicio de Cirugía General y del Aparato Digestivo, CSUR de Sarcomas y Tumores Músculo-Esqueléticos, Hospital Universitario Virgen del Rocío, 41013 Sevilla, Spain; danieldiaz@aecirujanos.es; 4Surgery Department, MD Anderson Cancer Center Madrid, 28033 Madrid, Spain; ilopezr@mdanderson.es; 5Department of Surgical Oncology, Hospital General Universitario Gregorio Marañon, Universidad Complutense de Madrid, 28040 Madrid, Spain; lozanon57@hotmail.com; 6Department of General Surgery and Surgical Oncology, Hospital Quironsalud Torrevieja, 03184 Torrevieja, Spain; paumozmoz@gmail.com; 7Peritoneal Carcinomatosis and Retroperitoneal Sarcomas Unit, San Juan de Dios Hospital, 14012 Córdoba, Spain; fcocris@gmail.com; 8Department of General Surgery, Hospital Clínico Universitario La Arrixaca, 30009 Murcia, Spain; vicenteolivaresripoll@gmail.com; 9Department of Surgery, Instituto Português de Oncologia de Lisboa, 1099-023 Lisbon, Portugal; hugovasques@sapo.pt; 10Hepatobiliary Surgery and Liver Transplant Unit, Gregorio Marañón University Hospital, 28007 Madrid, Spain; jmasencio@gmail.com; 11Department of Surgery, Faculty of Medicine, Universidad Complutense de Madrid, 28040 Madrid, Spain

**Keywords:** soft tissue sarcoma, delay, diagnosis, prognosis, litigation, non-planned resections, referral, red flag, conversion rate, detection rate, fast-track, pathway, 2WW, guideline

## Abstract

Diagnostic delay in soft tissue sarcomas is a common event with important prognostic implications. The authors review the available literature to obtain a general picture of its incidence, causes, and consequences. In addition, available clinical referral guidelines for early suspicion and referral have been proposed but have not provided the desired results. The authors review different proposed methods to improve results while avoiding a collapse of the system.

## 1. Introduction

Diagnostic delay in soft tissue sarcomas (STSs) is of paramount importance due to their prognostic implications and frequency [1]. Survival in sarcomas is determined by tumor grade, location, and size [2], and the latter is the only factor that can be modified by early diagnosis and is clearly associated with survival [1]. According to Davidson R, Chief Executive of Sarcoma UK: “Put simply, late diagnosis costs lives” [1]. Despite the importance of early diagnosis in STS, there are several barriers that make it difficult to achieve. First, STSs are very rare, with 1 in 100 soft tissue tumors examined by a pathologist found to be malignant. In the UK, the National Cancer Registration and Analysis Service (NCRAS) calculated an average of 3298 STSs diagnosed each year between 2013 and 2017, for an age-standardized ratio of 78.3 per million [3]. On average, general practitioners (GPs) will see about 3000 patients every year, i.e., only three benign soft tissue tumors per year and only one STS in 24 years, i.e., about two to three in his working life [4]. Second, STSs have a wide range of presentations in terms of location, size, age, etc. Third, the symptoms of an STS are no different from those of its benign counterparts [1]. Fourth, there are no tests or biomarkers prior to the onset of symptoms. Fifth, there is little or no awareness of its existence, which, together with the inexperience and lack of knowledge of the population and health care professionals, makes an early diagnosis difficult, if not impossible [1]. The dominant paradigm in sarcoma management is to gain access to new treatments in order to achieve better survival. From a global or systemic rather than individual patient perspective, the gain in survival can be achieved by minimizing the time from diagnosis to treatment [5].

The management and prevention of delay in diagnosis and treatment of STSs is also extremely complex due to several factors [6]. First, each sarcoma team in each health system has its own issues. Thus, although most of the causes are common, interventions need to be individualized [6]. Second, promoting early referral requires a minimum level of knowledge about STS and the “red flag” symptoms/signs that can help identify them, a basic knowledge that needs to be shared among patients, the general population, general practitioners (GPs), and non-sarcoma specialists [1]. Third, it is necessary to implement a clear and well-known referral guide that is linked to a simple and agile pathway [7]. Fifth, there is always a risk of saturation of the system due to an excess of referrals, making it imperative to establish several filters to ensure that the sarcoma multidisciplinary team (MDT) will only attend STSs without losing potential STSs [3,8].

## 2. Theoretical Framework in Diagnostic Delay

The study and analysis of diagnostic delay is extremely complex: diagnostic pathways are non-linear and strongly influenced by the health care system under consideration [9,10,11]. In addition, published studies often have poorly defined measurements and very different methodological approaches with little theoretical basis and a wide range of measurement points and intervals depending on data availability [9,10,11,12]. All this makes their results and conclusions difficult to interpret and compare. To further complicate the situation, different theoretical/methodological models have been developed regarding the definition and measurement of time points and intervals in early diagnosis research. These models facilitate the description of the events and processes that lead from symptoms to diagnosis and make it possible to identify where and why delays occur. Among others, we highlight the “Danish model” described by Olesen et al. in 2009 [13] and the “Pathway to Treatment model”, a 2011 version by Walter et al. [14] of the “Andersen model of Total Patient Delay” from 1995 [15,16,17]. Each model has its own advantages and disadvantages. The “Danish model” [13] categorizes the delay according to where it occurred, primary or secondary health care, and who is responsible (the patient, the doctor, or the system). It is a very simple model that is easy to use with well-defined points and time intervals. In contrast, the model developed by Walter [14] is much more complex, since the model includes not only key time points, defined as events, and time intervals between them, but also processes that may lead to the next event as well as contributing factors that influence processes, events, and intervals. Other models are more or less variants of the models described, most of which place emphasis on key points and time intervals (denoted T). Their complexity depends on the number of key points considered and, therefore, on the number of intervals considered. The model described by Ramos-Pascua et al. [18] considered only four time intervals (from T0 to T3). However, models such as the one described by Neal et al. in 2015 [19] considered 15, and the model of Tope et al. in 2023 described 33 [11]. Despite this, published studies differ substantially in study design, including different definitions of delay, measurements of delay, cofactors considered for delay, methods of analysis, and reporting of study results. In addition, no study has estimated and defined intervals of delay according to their associated non-medical and medical factors.

In 2009, an international Consensus Working Group (CWG) was convened with the aim of improving the quality and consistency of studies of diagnostic delay in symptomatic cancer, formulating definitions and key time points, and providing methodological recommendations for researchers [20,21]. The work was commissioned by Cancer Research UK and the Department of Health in England and carried out under the auspices of the Cancer and Primary Care Research International (Ca-PRI) Network (https://usher.ed.ac.uk/cancer-primary-care-research-international-network). The CWG concluded that, reviewing the literature available, the definition and measurement of key time points and intervals is poor, and there is very little guidance for researchers in designing studies. In addition, few studies explicitly used a theoretical framework to underpin definitions and diagnostic intervals. Finally, there was a lack of transparency and precision in the methods and instruments used in early diagnosis research. With all these findings in mind, the CWG eventually developed the “Aarhus Statement” (Table 1) and a checklist for early diagnosis researchers.

## 3. Elements and Time Intervals: What Is “Delay”?

It is important to identify the main actors involved in the diagnostic process, the time intervals considered, and the temporal distribution of the delay according to the available studies. Most of the studies identify the main actors of the delay as the patient, the family and the general population, GPs, the primary care (PC) nurse and physiotherapists (all included in PC), the general surgeon or non-specialized orthopedic surgeon (secondary care), and finally the sarcoma specialist, who is usually part of a multidisciplinary care team (MDT) in a tertiary hospital [21]. Considering these actors, we can define the following diagnostic time intervals: Patient or T0, primary care (T1), local hospital (untrained specialist) or T2, and sarcoma specialist center (T3) [18]. Of interest is the definition of preclinical tumor time or Tx. It is a confounding factor at T0, which cannot be considered as a “diagnostic delay” but it has an important value in deep sarcomas such as retroperitoneal. It is important to emphasize that there is no specific sarcoma model to help professionals identify areas of delay and therefore improvement. On the other hand, a model as such is not useful because each sarcoma unit is part of a specific health system that serves a specific population with its own characteristics. A general model could be useful, but always with modifications and adaptations according to the unit in question. In Figure 1, we present a very simplified and schematic sarcoma model in which we can identify the actors involved in the diagnostic process and the main time intervals considered. This model is quite simple and needs to be modified according to the health system and sarcoma group considered [21].

A commonly used measure in the literature is the “Time To Treatment Initiation” (TTI), which is usually defined as the time from first symptom to diagnosis or patient-related TTI (diagnostic delay) or, much more frequently, the time from diagnosis to treatment initiation or physician/hospital-related TTI (therapeutic delay). According to Schärer et al. [22], the TTI is defined as the time interval between histological diagnosis and surgery/radiotherapy (RT)/chemotherapy (CT) or the time interval between unplanned surgery and planned surgery/RT/CT. The TTI is considered as an important quality indicator in oncology, and its prolongation is associated with lower survival rates and increased anxiety and emotional distress in patients. On the other hand, TTI allows to know what the therapeutic process is like and how it unfolds. It favors the empowerment of the patient and the strengthening of the system–patient relationship, allowing evaluating the efficiency of the multidisciplinary team.

When can we consider a delay as unacceptable? There is no cut-off point or consensus that distinguishes between reasonable and unreasonable diagnostic delay. It is assumed that it should be as short as possible, with desirable times being communicated. The last version (2012) of the Danish CPP [23,24,25] recommends the following timescales for STSs: 13 calendar days from first appointment in the tumor center to decision on treatment; 14 calendar days (surgery), 15 calendar days (RT), and 11 calendar days (CT) from therapeutic decision to start of treatment; and 35–47 calendar days (surgery), 36–48 calendar days (RT), and 32–44 calendar days (CT) from date of referral to start of treatment. Clark, in 2005 [26], recommended 3 months as the maximum time interval from the first presentation to the first medical visit. Finally, Brouns, in 2003 [27], recommend 1 month as the maximum patient delay (from first symptom to first medical visit) and 1 month as the maximum medical delay (from first visit to diagnosis).

## 4. Extent of the Problem

In Table 2a, we describe the multi-institutional series representing the intervals of delay and their values [28,29,30,31,32,33,34]. It is noteworthy that only six series describe these data, which is very surprising considering the importance of this issue. Moreover, of these six series, one series includes only 72 sarcoma patients, and another series includes 182, which is not enough to draw solid conclusions. On the other hand, the systematic review by Soomers et al. [32] includes studies from 1945 to present, a too wide range that obstacles its conclusions, while the rest of the series are from the 2000s. In general, we observed objectively longer delays in all areas, with the patient delay being the longest of all. When analyzing specifically the patient interval, we found that in the Sarcoma UK 2020 survey (SUK20S), 25% of patients delayed their first medical consultation >3 months [33]. In contrast, in the SURVSARC study [30,31], this delay increases to 36%. Similarly, the diagnostic interval is >3 months in 47% of patients in the SUK20S vs. 28% in the SURVSARC study. The fact that the SUK20S was conducted only among patients and their relatives may explain these differences. Finally, it should be noted that it is rather difficult to compare the series because of the different definitions of the intervals, the unit of measurement (days, weeks or percentage of delay), and the type of sarcoma considered. In addition, the series have significant heterogeneity with a very wide range for all the data observed, probably related to the fact that each series belongs to a different health system with a different population in terms of socio-cultural and economic factors. In our opinion, these data should only be used as a guide to our own data, always with the aim of improvement. We encourage the sarcoma units to carry out this type of analysis to know the real data and identify areas for improvement.

In Table 2b, we have described single-institution series showing intervals of delay [18,27,35,36,37,38,39]. Again, the number of series is small (only eight series) with a small number of patients per series (around 100). The limitations we observe in the series described in Table 3 are the same here: high heterogeneity, wide range of data, and poor definition of time intervals. Again, patient delay is the longest interval of all.

Table 3 shows the series describing the TTI [22,40,41,42]. Again, the number of series is very small. Two of them are based on the US National Cancer Database (NCDB) with 23,786 and 41,529 patients, respectively [41,42]. The other two are based on databases of two sarcoma networks (Switzerland and New Zealand) and include only 266 and 223 patients respectively [22,40]. The range of mean TTI is 14–42 days, with the observation that these values increase with time. Curtis, in 2018 (42), observed an increase in TTI times in 30% from 2004 to 2013.

## 5. What Is the Cause of Delay?

The answer is quite complex. It is imperative to group the causes according to their position in the diagnostic pathway, without forgetting that all causes are interrelated, as it is impossible to isolate one from the others. In this review, we have grouped the causes (Table 4) according to the main actor involved (tumor, patient, general population, PC/untrained specialists, health system, and sarcoma team). Others, such as Bourgeois et al. [43], argue that barriers are multidimensional and prefer to distinguish among micro-, meso- and macro-level barriers. It is important to remember that all barriers interact and intersect, regardless of the classification used.

### 5.1. Tumor Related Causes

Multiple causes are related to the tumor being observed that the average delay in diagnosis of a sarcoma is at least 7 months longer than for any other benign condition [36]. Size is an important factor as larger tumors are associated with shorter time intervals, not only due to its size itself, but also because as the tumor grows, the patient develops symptoms such as pain, which could favor an early diagnosis. Not all authors report this. Dyrop et al. [36] observed longer patient intervals for tumors >5 cm. It has been reported by other authors that the presence of pain increases the interval due to its association with benign conditions [7,44]. Tumors located in the upper extremity have a smaller size (7.3 mm) than those in the lower extremity (10.2 mm) because they are easier to detect [36]. In the same regard, tumors located under the superficial fascia (deep) are diagnosed 10 months earlier than superficial STSs due to the lack of suspicion in superficial tumors as they use to be considered benign [36]. In 2023, Elyes et al. [29] found that deep STSs had shorter intervals than superficial STSs: patient interval, 8.3 weeks vs. 20.7 weeks; diagnostic interval, 6.9 weeks vs. 5.7 weeks; PC interval, 0.4 weeks vs. 0 weeks; secondary care interval, 3.9 weeks vs. 8.1 weeks; and finally, tertiary care interval, 1.6 weeks vs. 0.9 weeks. The total treatment interval was 20.9 weeks for deep STSs vs. 34.8 weeks for superficial STSs. It is clear that anatomical location has a definitive influence on delay. Skin sarcomas such as Kaposi sarcoma (KS) or dermatofibrosarcoma (DFS) are easily detected and treated, whereas retroperitoneal sarcomas are usually diagnosed late and therefore with a large mean tumor size. Other factors to be considered include tumor grade, histology, and stage. High-grade (HG) STSs are diagnosed 42 months earlier than low-grade (LG) STSs because HG STSs have more severe symptoms and are more likely to be selected for fast-track referral [33,43]. In contrast, the diagnostic interval used to be longer, a finding associated with the need for longer and more complex preoperative investigations [36,45]. Nandra et al. [45] found significant differences in patient interval times according to histology: lipo- and leiomyosarcomas had an average of 26–28 weeks before diagnosis compared to 52 weeks for synovial sarcoma. Despite this, data are confusing and sparse due to the existence of >160 histological types/subtypes. On the other hand, slow growing sarcomas, with potentially “benign” histology, usually LG STSs, have a long pre-diagnostic history, associated with relatively good prognosis. In contrast, rapidly progressing tumors, normally HG STSs with very aggressive histology, may experience less profound impact on diagnostic delay [45]. Finally, advanced stage is a cause of greater delay in treatment initiation and is also a cause of treatment discontinuation due to the need for symptom control prior to treatment and the frequent need to hospitalize patients [43].

### 5.2. Patient-Related Causes

Several barriers for early diagnosis can be related to the patient [43,46,47]. It must be emphasized that the patient-related time delay is the longest of all intervals. According to Brouns et al. [27], 47% of patients take >1 month from symptom onset to consultation. It is possible to identify several factors associated with this delay, but the most important is low or no awareness of the disease and its consequences, usually associated with a lack of “public/family awareness” of sarcoma. Related to this is the fact that most patients misattribute symptoms to benign conditions or underestimate the severity of symptoms. From an epidemiological point of view, several considerations must be highlighted. Diabetics and smokers have longer delays because they pay less attention to their health status. Mental health conditions such as depression, anxiety or psychiatric illness, and/or substance use are considered barriers to treatment discontinuation [43]. Children, adolescents, and young adults used to have longer delays, which are related to the low incidence of malignant pathology in this age group and the attribution of symptoms to their lifestyle [43,46,47,48]. Genital tumors used to be diagnosed later (as was the case with breast cancer years ago) because of patient reluctance to seek medical help [46].

Through a systematic review, Syros et al. [47] identified at least three distinct groups of causes of patient delay. The first group is the socio-economic status. It is considered a fundamental barrier to accessing care and includes issues such as the availability of private insurance or income status. In the USA, its presence is associated with poor access to treatment, poor adherence, and greater delay to receive it [43,46]. Miller et al. [49] observed that patients with Medicaid insurance have lower survival rates in various types of sarcomas compared to those with private insurance (HR: 1.18), an observation almost identical to that of Smartt et al. [50], with a reported HR of 1.2 in STSs. Similarly, Jang et al. [51] observed higher mortality rates in patients with Medicaid insurance (HR: 1.28). In contrast, uninsured patients had worse outcomes compared to patients with non-Medicaid insurance (HR: 1.6). Finally, Malik et al. [52] demonstrated the positive effect of insurance legislation and Medicaid expansion, with an increase in early-stage cancers and a decrease in late-stage cancers. With regard to economic factors, it has been observed that lower income classifications in South East Asian countries are associated with higher rates of treatment refusal/abandonment [51]. In addition, being in the lowest quartile of socioeconomic status is a predictor of lower 5-year survival (HR: 1.23) [49]. Unstable housing, usually associated with poor economic status, has been identified as a barrier to cancer treatment due to inadequate storage of medication, risk of theft of belongings, and poor living conditions, all of which are considered incompatible with managing the side effects of chemotherapy or recovering from surgery [43]. Finally, a report from the NCRAS in the UK [3] showed that patients living in the fifth deprivation quintile were 22% more likely to die of their disease than those in the reference group, a finding that the authors linked to poorer access to specialist services.

The second group is the “distant decay” or geographical barriers, which have an important impact on sarcoma care, as oncological outcomes deteriorate the further patients live from a referral center [53]. However, the influence of distance is small when compared with clinical factors. Patients living further away from reference centers were less likely to have early access to specialized diagnosis because those who are going to be treated in a distant site incur financial expenses and some other costs, such as travel time and expense, possible need for lodging and food, reliable transportation, and additional time away from childcare responsibilities [43,53]. According to several authors living more than 2 h [54] or more than 100 miles from a specialist sarcoma center [55] was associated with worse outcomes. Similarly, living in a rural area or in a deprived area was initially associated with poorer survival (HR: 1.23), but this association was not significant when adjusted for clinical variables (HR: 1.03) [55].

Socio-cultural barriers are the third group of barriers comprising factors such as lack of understanding of what sarcoma is and anxiety about treatment plans [56]. Being single has been observed to be associated with HG STSs, less RT, and less surgery and has also been identified as an independent predictor of sarcoma-specific death [57,58]. With regard to ethnics and racial disparities, Alamanda et al. [57,58] found that African American race was associated with larger tumor size, less RT, fewer surgeries, and a higher number of deaths. With regard to the effect of educational status, it has been observed that patients with low levels of education have longer patients’ intervals due to financial constraints, cultural and religious beliefs, and lack of access to health care facilities [43,59]. Finally, immigrants, who make up a significant proportion of the population in several countries, underutilize health care systems, particularly cancer screening programs, due to factors such as language barriers, education levels, or low socioeconomic status [43,53]. It has been observed that communication challenges, including understanding written and verbal communication are a barrier to care. This is especially true in those with low economic status, which is associated with treatment delay and non-adherence or perception by health care providers that patients are less engaged with the treatment plan [41,53].

Finally, it is important to emphasize that all of these factors tend to occur simultaneously, a situation usually defined as “compounding” or “multiplicative” [43]. That is, those who experience multiple barriers are less likely to receive treatment. Authors such as Costas-Muniz et al. [60] have shown that the presence of >4 unmet socioeconomic and supportive care needs is associated with missed RT/CT appointments. According to Smartt et al. [50], Medicaid patients constitute a socioeconomically disadvantaged subgroup of the general population likely to be disabled, with multiple physical or psychiatric comorbidities usually with substantial financial barriers.

### 5.3. General Population-Related Causes

The main reason for the delay in the general population is a lack of “public awareness”. According to the SUK20S [33], only one-fifth (21%) of families had ever heard of sarcoma. In addition, only half (49%) knew that sarcoma is a type of cancer. Perhaps less impressive, but still alarming, is the fact that 67% of relatives found out about sarcoma online, rather than from their doctor. However, these results must be interpreted with caution due to the presence of geographical variability, cultural issues, and other factors that ultimately influence the population’s ultimate knowledge and awareness of sarcoma.

### 5.4. Primary Care-Related Causes

PC (GPs, nurses, and physiotherapists) act as gatekeepers for sarcoma, with 90% of patients having their first consultation here, and it is here where many patients face their first barriers to diagnosis [43,46,47]. For many, this is one of the most common causes of delay in diagnosis, in most cases due to a lack of initial suspicion. Indeed, some studies have shown that when GPs consider the possibility of sarcoma, the delay in diagnosis is reduced by around 16 months. Despite this, among PCs, it is possible to identify two main causes of delay in diagnosis. The first is lack of suspicion because sarcoma is not thought to be present. Indeed, some authors suggest that STS is suspected in only 33% of cases in primary care. The second is lack of knowledge because the signs/symptoms of STSs are not well defined and the red flags are not known. Finally, and clearly related to lack of knowledge, is misinterpretation of imaging and/or biopsy data. In summary, a combination of inexperience and lack of knowledge in this pathology leads to poor management, with patients being observed, resected (unplanned resection), or poorly and late referred instead of promptly referred [43,46,47].

Are PCs really unaware of sarcomas? Is the level of knowledge about this pathology really low in PCs? Fossum et al. [61], in 2020, tried to answer these questions using a survey of PCs. The authors observed that, among physicians, familiarity with guidelines reached only 3 points on a scale of 10 (interquartile range: 2–5) with a self-confidence in the management of a suspected sarcoma of 3/10 (interquartile range: 3–5). These data are closely related to the fact that the average number of STS patients during a physician’s career was 2.2. In addition, 32% of doctors had never seen a sarcoma patient. Previous studies [62] have suggested that sarcoma diagnosis may be delayed due to assumptions of non-malignancy among PCs, as these tumors are much more likely to be benign than sarcoma. In this sense, it is possible to consider knowledge as a key factor in PC-related delay, but it is only one of many. It is significant that the authors found no differences in sarcoma case management between GPs and nurses, or with years of experience, previous experience with sarcomas, knowledge of clinical guidelines, or postgraduate attendance at sarcoma conferences. In fact, PC physicians identified several other factors contributing to delayed diagnosis, including patient delay, time available for consultation based on clinical time constraints, insurance restrictions, delays in obtaining imaging studies, and misinterpretation of imaging studies [61].

A key marker of diagnostic timeliness is the number of PC consultations prior to referral. As the number of consultations increases, so does the delay. It was observed that the delay reached 1 month with three PC consultations, 1.5 months with four PC consultations, and 3 months with more than five PC consultations [63,64]. In this sense, a pathology is considered “difficult to suspect” if >30% of patients have >3 PC consultations (as happens with sarcomas), whereas it is considered “easy to suspect” if <15% have >3 PC consultations [63,64]. This is the “sarcoma signature” or “symptom signature” [65,66]. In 2016, Mendonca et al. [65] observed that 31.9% of sarcoma patients consulted in PC >3 times. Others, such as Dyrop et al. in 2016 [36], reported a mean number of PC visits of 1.6 + 1.1. The SUK20S [33] reported that 35% of the patients visited GPs at least three times. In a recent retrospective study by Rafiq et al. [67] in 2024 of 377 STS patients, the authors examined PC utilization over time before a sarcoma diagnosis. The authors found that 84% of STS patients visited a GP in the 6 months prior to diagnosis but only 36% had an imaging request. Notably, repeated GP visits were common, with 52% of STS patients having visited their GP at least ≥4 times in the 6 months before diagnosis, with a median of 3–4 GP visits in the 6 months before diagnosis and 6 GP visits in the year before diagnosis. The rate of GP imaging requests per month increased progressively from 6 months prior to sarcoma diagnosis, peaking at 8 times the baseline rate immediately prior to diagnosis. This gradual and initial increase was followed by a more rapid increase in the 3 months before cancer diagnosis. The SUK20S [33] found similar results, with 42% of GPs referring patients for additional tests (laboratory and/or imaging) and 18% being referred to a specialist. This means that around 40% of GPs were not guiding patients in the right direction for a prompt and accurate diagnosis. Finally, in a recent study, Holthuis et al. [68] report that the mean monthly number of consultations was statistically higher from 4 months prior to diagnosis, with the highest number of consultations occurring in the last month. Accordingly, it is possible to identify a “diagnostic window”, a period of more or less 6 months prior to diagnosis, in which there are potential opportunities to speed up the diagnosis of sarcoma in some patients [67].

In summary, the most important factor associated with delayed diagnosis is lack of awareness: “If you do not think about sarcoma, you will never diagnose a sarcoma”. In fact, we can describe a sequence of errors starting with a lack of suspicion, i.e., an initial incorrect diagnostic hypothesis leading to an incorrect diagnostic/treatment cascade (Figure 2).

### 5.5. Health System-Related Causes

Several causes related to the health system can be identified [43,46,47]: lack of agile and simple communication circuits between PCs or non-specialist surgeons and the MDT; lack of fluid relationships between professionals (personal issues and professional “itching” still block the way to diagnosis in several diseases such as sarcoma); lack of implementation of clinical guidelines, a factor related to the low compliance observed; the area attended by the health system considered, as residents in rural areas and/or living far from the tertiary hospital where the sarcoma unit is located use to have worse results; and finally, the lack of continuity of care and fragmentation between cancer care services, usually referred to as “fractured” or “soiled” cancer care system [43].

Finally, it is interesting to highlight other factors. First, general orthopedic surgeons refer sarcoma patients earlier and better than general surgeons. A finding related to the fact that training in sarcomas is applied in 84% of trauma residents vs. 35% in general surgery [69]. Second, the role of the economic characteristic of the health system considered, i.e., whether it is public or private, is important, and this is closely related to the country considered [43,46]. Prior to the expansion of Medicaid in 2014, the mortality of uninsured sarcoma patients in the US was 28% higher than that of their insured counterparts. Since Medicaid expansion, we have seen a decrease in sarcoma mortality, demonstrating the direct relationship between increased access to care and survivorship. This relationship may be partially explained by the fact that insurance status has been shown to be a positive predictor of clinic attendance, which may prevent delays in care and advanced stage presentation of sarcoma [43,46].

### 5.6. Sarcoma Team-Related Causes

The complex organization of the MDT is the reason for some barriers to care. In this sense, TTI depends on several elements of the MDT and not just on surgeons, as is commonly thought. The MDT is composed of different professionals such as radiologists, pathologists, and others. Seinen et al. [38] analyzed the delay in the management of retroperitoneal sarcomas and tried to identify which elements of the MDT are responsible for the delay. The author found that the median pathology lead time was 22 days, with some of the longest delays caused by inconclusive pathology reports. The radiology lead time was a median of 36 days, with requests for different investigations rather than coordinated investigations likely to have contributed to the delay. Other causes such as investigations carried out in another hospital also contribute to the delay.

TTI, as mentioned earlier, is an interval that describes the time between diagnosis and treatment and is usually almost exclusively related to health system issues, specifically in the MDT. As noted above, this interval is getting longer every year, perhaps because diagnostic and therapeutic techniques are much more complex and because of the higher volume of patients seen at referral centers. The literature review identified two studies that addressed and identified factors associated with prolongation of the TTI. In 2017, Curtis et al. [42] analyzed 41,529 patients from the NCDB between 2004 and 2013, identifying the following as good prognostic factors (expressed as incidence rate ratio): high socioeconomic status (0.95), trunk location (0.94), HG STSs (0.92), and stage IV (0.92). On the other hand, the same authors identified risk factors for prolonged TTI: no private insurance (1.13–1.18), being treated in an academic hospital (1.23), being referred from another center (1.76), and being treated with RT/CT (1.4–1.53). Similar to Curtis, in 2021, Ogura et al. [41] analyzed 23,786 patients from the NCBD between 2004 and 2015. The authors identified the following as good prognostic factors: operation at a non-academic hospital (0.64–0.86), a high level of education (0.9–0.95), and HG STSs (0.85–0.88). On the other hand, the same authors identify risk factors for prolonged TTI: female (1.04), non-white race (1.03–1.12), no private insurance (1.12–1.34), high economic level (1.05–1.07), high comorbidity (1.06–1.15), STSs located at the trunk/lower extremities (1.03–1.15), and referral from another center (1.62). It is clear that many of these results cannot be extrapolated to every country/health system all over the world, as they have been obtained from US patients, and some of them must be interpreted with caution. The only point of interest, in our opinion, is that the most important risk factor identified by both studies is referral from another center. This issue highlights the problem that we find specifically in health care systems, i.e., the existence (or not) of a fluid and well-designed referral route. It is possible to identify several other factors that significantly influence the TTI. Schärer et al. in 2023 [22] identified tumor location (deep or superficial) and grade as important factors. The authors showed that superficial STSs tend to be treated late, with a median TTI of 42 (27–71) days versus 28 (18–48) days for deep STSs, with the same differences observed when STSs were malignant (42 days vs. 22 days) or intermediate grade (43 days vs. 54 days).

## 6. Consequences of a Diagnostic/Treatment Delay

### 6.1. Prognostic Impact of a Late Referral

Several studies have found a strong association between delay in diagnosis and prognosis in cancers in general, but data on sarcomas are scarce and confusing [70]. In a recent systematic review and meta-analysis by Hanna et al. [5] on seven major cancers (34 studies with 17 indications), delay in surgical treatment was associated with a 1.06–1.08 (6–8%) increase in the risk of death per 4 weeks of delay. In the same way, another recent systematic review by Neal et al. [19] with 117 articles and 209 trials covering many types of cancer (including sarcoma) found an association between shorter time to diagnosis and more favorable outcome. In the specific case of sarcomas, the review analyzed five trials, of which three showed a positive association with survival and two showed a negative association with survival and stage. The systematic review and meta-analysis by Soomers et al. [32], which focused only on sarcomas and included 62 studies (36 on STS) from 1946 to 2020, also found conflicting results. The authors reported that only 10 retrospective studies analyzed the relationship between prognosis and treatment delay. In five studies, the delay had no effect on survival. Only one trial of STSs showed improved survival with a longer total interval (26 vs. 20 weeks). Finally, three trials showed improved survival with shorter total intervals.

These conflicting results are in contrast to what we can expect if the diagnosis and treatment of a malignancy is delayed [32]. In the specific case of sarcomas, early diagnosis should be associated with a reduced extent of surgery and increased survival. In contrast, late diagnosis should be associated with larger tumors, more extensive surgery, poorer prognosis, and an increased risk of metastasis and amputation. The absence of correlation between delay and prognosis is termed the “waiting time paradox”; it is also defined as a phenomenon where both short and long waiting times before diagnosis result in decreased survival [32]. The reasons for this finding are varied and depend on the characteristics of the malignant tumor analyzed [71]. In the case of sarcomas, if we consider only the most aggressive tumors as HG STSs, we see that they are treated more quickly, reducing the delay, but with the same poor prognosis. On the other hand, LG STSs could benefit from a better pre-therapeutic assessment, increasing the delay, but with a better outcome related to the biology of the tumor. In fact, for some cancers, although the delay is highly undesirable, the biology of the cancer may be more important than the delay in diagnosis [70]. Rougraff et al. [71] listed several conditions of a cancer where a temporary delay affects prognosis. The cancer must be potentially curable, which is not the case with HG STSs, but also potentially fatal, since almost all cases of LG STSs are cured regardless of the waiting period. The development of metastases must occur after the cancer is diagnosed. If metastases develop in an unpredictable way, as happens with HG STSs, or long before the primary cancer is diagnosed, the delay will not affect the prognosis. Of course, metastases must mean a worse prognosis, as is the case with sarcomas. Increasing tumor size must be negatively associated with prognosis, a common finding in sarcomas. Finally, patients must have sufficient life expectancy after diagnosis to avoid death from other causes, such as age. Because of the relatively young age of sarcoma patients, this factor is not taken into account.

In Table 5 [37,40,41,45,72,73,74,75,76,77,78], we described all series that addressed the relationship between prognosis and delay in diagnosis/treatment in sarcomas. We identified 13 studies [32,37,40,42,45,71,78], of which one was a systematic review [32]. The number of patients analyzed in each study is very small, ranging from 73 [77] to 624 [71], except for two studies that reviewed the US NCDB with 23,786 and 8648 patients, respectively [41,73]. As we observe, there is an important heterogeneity in terms of the period of analysis, the histology of the sarcomas included, and, most importantly, the interval of time analyzed. The results of the series show the presence of the waiting time paradox in almost 50% of the cases. These findings may explain why it has been difficult to prove that long waiting times worsen the prognosis for sarcoma patients.

### 6.2. Non-Planned Surgical Resection of Sarcomas

Due to the well-known lack of awareness of the possibility of sarcoma in soft tissue tumors, delays in diagnosis/treatment are common. In parallel, there is a high incidence of inadvertent excision of lesions that turn out to be sarcomas. First described by Giuliano and Eilber in 1985 [79,80,81], this is known as the “whoops” procedure, inadvertent excision or non-planned resections (NPRs), as the surgeon is surprised when the pathologist reports that the lesion he thought was benign is in fact malignant and is partially resected.

The characteristics of the lesions resected in an unplanned way have been well described. In a recent review by Larios et al. [82], with 43 studies and 16,946 extremity/trunk STSs, the authors found that 35.5% were NPRs, of which 60% had residual tumor in the surgical bed. In order of frequency, NPRs were located in the lower extremities, followed by the upper extremities and trunk. Most of them were LG STSs, including superficial and small tumors. Belzarena et al. [83] report similar results in a recent retrospective case–control study. The authors observed a 23.4% incidence of NPRs, most of which were initially thought to be simple lipomas. NPRs were more common in the upper extremities, less prone to postoperative infection or dehiscence, with higher rates of amputation, with a higher rate of positive margins, and were usually superficial and small in size. It is interesting to note that 46.7% and 20.3–23.4% of NPRs were performed by general surgeons and non-specialized orthopedic surgeons, respectively. In fact, orthopedic surgeons order imaging studies for soft tissue tumors in 47.3% of cases compared to 18.2% in other specialties. Venkatesan et al. [84] reported in 2012 a wide range of specialties performing NPRs, with general surgery being the most common (38%), followed by non-specialist orthopedic surgeons (14%), GPs (14%), and plastic surgeons (14%).

These findings are not coincidental. Small and superficial tumors, usually LG sarcomas, in easily accessible areas of the body are often misdiagnosed as benign by GPs, general surgeons, or others, and resected, usually under local anesthesia, without any further complementary investigation. Belzarena et al. [83] analyzed, using an Ishikawa diagram or root cause analysis (RCA), the obstacles encountered along the time line leading to NPRs. They found five main obstacles, each with its own causes: 1. The physician’s assumption of a benign diagnosis (45.8%), which is usually related to the fact that benign conditions are far more common than malignant ones, the presence of a history of trauma, and the presence of non-specific symptoms; 2. Inappropriate imaging studies ordered (76.6%), which is related to availability or lack of MRI and insurance authorization and lack of awareness; 3. Incorrect imaging report (65.2%), which is related to inappropriate imaging studies and lack of awareness; 4. Failure to request biopsy (88.6%), again related to sarcoma unawareness and incorrect imaging report; and 5. Incorrect biopsy report (75%), which is related to the fact that sarcoma diagnosis is difficult and elusive and sarcoma unawareness. Other authors such as Venkatesan et al. [84] report similar findings. In a study of 42 NPRs in 216 patients (19.4%), the authors found that the initial clinical diagnosis was benign in 83.3%. Adequate preoperative imaging was performed in only 25% of cases, and preoperative biopsy was performed in 2.3%. Finally, resection was performed by an inadequate incision in 33.3%. Finally, in a recent Swiss Sarcoma Group study, Schärer et al. [85] found that NPRs are more common in fragmented pathways, which increases the risk of LR but does not affect survival, as STSs tend to be smaller, superficial, LG STSs with less aggressive histology. Conversely, patients undergoing planned resections in comprehensive care pathways are more likely to present with larger, HG tumors at a more advanced stage, increasing the risk of distant metastases and negatively impacting survival.

In conclusion, there are several reasons to explain why NPRs are still performed, but lack of knowledge, poor training, and lack of awareness are the key factors. It is thought that the better the education of GPs and general surgeons is, the lower the proportion of NPRs. The preoperative suspicion (“to think in sarcoma”) plus a complete evaluation of the surgical specimen, namely, the biopsy, are the keys to avoid NPRs.

### 6.3. Medico-Legal Impact of Late Referral

Medical malpractice litigation in the United States is a major concern for physicians with an estimated global malpractice claim rate of 7% [86]. These data contrast with that observed in high-risk specialties, such as general surgery and orthopedics, where 15% of physicians will face a malpractice claim [86,87]. For this reason, more than 90% of physicians in high-risk specialties admitted to practicing “defensive medicine”, a practice associated with an estimated USD 45 billion annual cost to the health care system [88]. Despite this, just six studies [89,90,91,92,93,94] analyzed the medico-legal impact of late referral in sarcomas (Table 6).

We observe that the number of series analyzing the legal implications of delayed diagnosis of sarcoma is extremely small, only four series [89,90,91,92]. Not surprisingly, they all come from the USA and UK, countries well known for their legal pressure on doctors. It should be noted that the total number of claims identified is low, around 400, perhaps because most claims do not go to court, being resolved by out-of-court settlements. The most common reason for medico-legal claims related to sarcoma care is delay in diagnosis, with GPs and non-oncology orthopedic surgeons being the most commonly named physicians in these claims. The most commonly reported injury as a result of the alleged delay in diagnosis was progression to metastatic disease. Finally, it should be noted that the average award is extremely high, particularly in the US (about USD 2,000,000) [95].

### 6.4. Psychological Impact of Late Referral

According to SUK20S [30], a delay in diagnosis adds to the high emotional burden that a diagnosis of sarcoma generally imposes on patients and relatives, usually in the form of anticipatory anxiety and uncertainty before the diagnosis and later consideration of whether the delay has affected their prognosis, although survival is a relatively long-term outcome. To these factors must be added the distress caused by the possible poor cosmetic appearance after surgery and how the delay has affected it. The same survey showed that 9 out of 10 patients said sarcoma diagnosis affected their mental health, the extent to which is severe in 25% of cases. By gender, 28% of women and 19% of men are severely affected. In terms of age, 45% of patients aged 16–24 years, 31% of patients aged 25–40 years, and 29% of patients aged 41–55 years were severely affected.

## 7. “Red Flag” Symptoms of STSs

The early diagnosis of STSs is based on clinical data because there is no clinical or blood test for diagnosis [6,96,97,98]. In this setting, the use of alarm symptoms or “red flags” by GPs helps to identify these patients [6]. The classic and conventional symptoms/signs associated with STSs are [96,97,98] tumor size, location in relation to the fascia (superficial/deep), growth, pain, and symptom duration. STSs usually present as indolent lumps that are occasionally painful due to pressure on adjacent structures. Benign soft tissue tumors tend to grow slowly, in contrast to aggressive tumors, which grow rapidly. STSs are typically large and localized deep to the fascia, making their detection difficult until they reach a large volume. Despite this, most patients with these symptoms do not have a STS because benign soft tissue tumors far outnumber STSs and they present with the same symptoms, making them very unspecific.

### 7.1. Size

It is the most important variable predicting survival and the only factor potentially modifiable through early diagnosis to improve prognosis: Pisters et al. [2], in 1996, observed a RR of 2.1 in STS > 5 cm vs. <5 cm. Kolovich et al. [99], in 2012, showed that an increase in size of 1 cm implies a RR of 1.058 of death with a decrease in survival of about 3–5%. Finally, Grimer et al. [100], in 2006, compared STS < 5 cm vs. >25 cm and observed a higher rate of metastases (3% vs. 18%) and a decrease in overall survival (OS) with HR values of 1 vs. 8.5, respectively. Since the 1980s, a cut-off point of 5 cm has been considered the threshold at which a soft tissue tumor is suspected of being malignant. However, several studies have lowered this to 4.3–4 cm. Nandra et al. [101], in 2015, developed the “golf ball” model. The authors found that 30% of soft tissue tumors <4.3 cm were malignant, compared with 59% of those >4.3 cm, with an OR of 3.65. With these data, the authors developed their model, which is not precise enough, but is easy to remember. Smolle et al. [41] found that size was the sign with the highest sensitivity and also the second best specificity of the four warning signs. Despite a decrease in specificity to 48%, the authors recommended changing the threshold from 5 cm to 4 cm, as an increase in sensitivity of up to 89% was also observed. Based in the previous data, it is recommended in the UK to suspect a potential STS when a tumor has a size > 4 cm. In other countries, such as Denmark, this change has not been established due to fears about how many extra referrals of patients with benign conditions this change would generate. Despite the established thresholds, we must always remember that even lesions < 2 cm may be malignant. Recently, Gassert et al. [102] reviewed 870 STSs < 5 cm and showed that 22.4% were <5 cm, 16.5% < 3 cm and 15% < 2 cm. These data are consistent with others reported by authors such as Pham [103] (21% < 2 cm), Obaid [104] (15% < 5 cm), and Datir [105] (10% < 5 cm).

### 7.2. Location

Location in relation to the fascia is considered a second-order clinical factor because of its average sensitivity and specificity, especially when compared with size, and because it has no predictive value in high-grade (HG) STS > 5 cm, where tumor size is much more important than depth. In a 2008 study by Datir et al. [105], 42% of malignant lesions were deep, but up to 29% of benign lesions were also deep. On the other hand, depth is difficult to assess, both for the patient and the clinician, and is strongly influenced by pain. Finally, it is important to remember that most STSs are deep. Brennan et al. [106], in 2014, after analyzing 10,000 patients, identified only 8% of STS > 5 cm that were superficial.

### 7.3. Tumor Growth

Tumor growth is a typical finding in STSs, as 2/3 of STSs have it compared to <50% of benign soft tissue tumors, with a sensitivity of around 75% and a specificity of around 51%. According to Nandra et al. [101], its presence has an OR of 5.53 for malignancy. However, there are important drawbacks to this finding, such as the need for an observation period (“wait and see”), which is clearly associated with a potential delay in diagnosis, and the fact that it is a very subjective factor based on patient judgement, as most patients will report that the tumor has grown once they have become aware of its presence. Tumor manipulation and imagined growth due to anxiety may also play a role.

### 7.4. Pain

Pain, originally included by the NICE (National Institute for Care and Care Excellence, London, UK) [107] as a clinical factor for referral to sarcoma units, it is now recommended to be removed. Additionally, the Lund Group (Sweden) [108] believes that it should not be included in referral guidelines, while the Danish CPP [24] includes pain as a warning symptom only for bone tumors. This is because it is a subjective clinical factor that is site dependent and also dependent on local irritation or compression. Moreover, it is a very common symptom, especially in PC, where it is usually associated with benign pathology. We must remember that some studies showed that up to 50% are painful. Finally, its PPV in the general population is very low. Considering all these data, it is easy to understand why authors such as Nandra et al. or Smolle et al. consider it as a second-grade clinical factor with very poor sensitivity and specificity [41,101]. In contrast with these data, Elyes et al. [109] reported in a recent work that pain accelerates the diagnostic process, while other symptoms as swelling or sensory disturbances delay it.

### 7.5. Duration of Symptoms

In 2006, Grimer et al. [110] analyzed 1460 STSs, classifying them according to the duration of symptoms (more or less than 6 months). The authors reported that those with a longer duration of symptoms had a smaller tumor size (10 vs. 11 cm), a higher proportion of low-grade (LG) STSs (33% vs. 20%), and a better prognosis with a HR of 0.998. The authors conclude that there is an inverse relationship between duration of symptoms and OS because aggressive STSs grow rapidly, reaching a large and alarming size and compressing surrounding structures, usually causing symptoms such as pain. All of this leads patients to seek medical attention, thus reducing the time to diagnosis [110,111].

### 7.6. Predictive Value of Alarm Symptoms

The value of these symptoms/signs depends on the prevalence of STSs. Shapley et al. [110] consider a PPV > 5% to be highly predictive of malignancy, but most of the symptoms/signs already discussed fall below 5% in PC. In 2015, Nandra et al. [101] analyzed 3018 soft tissue tumors, 48% of which were malignant. The authors identified increased size as the most sensitive factor (71.3%) and pain as the less sensitive (44.2%). The most specific was size > 4.3 cm (62.8%), and the less specific was deep location (29.9%). The OR for malignancy of each “red flag” finding was 3.65 for size > 4.3 cm; 1.3 for pain; 5.53 for growth; and 1.5 for deep location.

Ideally, a combination of symptoms and signs may help to identify potential STSs rather than using just one feature, as the highest PPV was seen for combinations of symptoms. Johnson et al. [111] in 2001 analyzed 526 patients, 275 of which were malignant. The authors found that the presence of three or four clinical features (“red flags”) was associated with malignancy in 81% and 86% of cases, respectively. They also identified tumor growth as the best indicator of malignancy and size < 5 cm as the best indicator of benignity. The authors determined the weight of evidence for or against malignancy for each of the symptoms/signs. The summed weight of evidence was then plotted on a graph, allowing the probability of a lump being malignant to be calculated. Increase in size has a weight for evidence of malignancy of 1.27, and the location deep to the fascia a weight of evidence against malignancy of −1.59. Nandra et al. [101] showed similar results. Tumor growth in a deep mass had a sensitivity of 93.4% but a very low specificity (36.1%). In the same regard, a growing mass > 4.3 cm has a PPV of 78.5%, with high sensitivity (89.4%) and specificity (63.6%). They also found that the greater the number of positive findings is, the greater the risk of malignancy: one finding, 30%; two findings, 40%; three findings, 60%; and four findings, 80%. The authors also constructed a Bayesian belief network that allowed them to identify first- and second-degree associates in relation to the outcome of interest (benign/malignant). First-degree associations were increasing size, age, size of lump, and duration of symptoms, while second-degree associations were depth, location, and presence of pain. Finally, the authors recommended that size >4.3 cm, increasing size, increasing age, and shorter duration of symptoms be used as “red flag” clinical features, always bearing in mind that the more features we find, the more likely it is to be malignant. Smolle et al. [41] came to almost the same conclusions. Specifically, the risk of malignancy increases as the number of findings increases: one criterion, 18%; two criteria, 33%; three criteria, 62%; and four criteria, 43%. If the lump is deep, growing, and >5 cm when examined, the risk of malignancy rises to 77%.

Sarcoma patients presenting with alarm symptoms had shorter waiting times to diagnosis, a finding also observed in other cancers. In contrast, those presenting without alarm symptoms have been shown to have longer diagnostic delays and higher mortality. Hussein et al. [112] observed a correlation between the presence of “red flag” symptoms and delay (clinician-specialist): 5.4% had no criteria and a delay of 33.15 months vs. 94.6% with any criteria and a delay of 19.86 months. When 3 months was chosen as a realistic cut-off for delay, 36% of patients had a delay of <3 months, of whom 97.2% had any clinical alarm criteria. In contrast, 64% of patients had a delay of >3 months, of whom 93.3% had some alarm criteria.

### 7.7. PC and Alarm Symptoms

George et al. [109] in 2012 found that the most common reason for consulting a GP was a painless lump (71%), of which 35% cited an increase in size as a reason for immediate consultation. In fact, the author found that increasing size was the only factor associated with longer patient delay. These data are almost identical to those observed in the Sarcoma UK 2020 Survey [33], where the most common reason for consultation was the presence of a painless lump (34%), followed by a lump increasing in size (26%), and a painful lump (17%). Dyrop [36] analyzed the same question in 2016 identifying pain (14.7%), the need for a diagnosis (11.8%), family pressure (9.8%), and the presence of a lump (8.8%) as the most frequent reasons for consultation.

An issue of interest is the definition and identification of alarm symptoms/signs in PC, which varies considerably from those obtained from specialist units. In a 2012 study, George et al. [109] compared the presentation and symptoms between PC and specialist units and found important differences. At first presentation to PC, 17% of STS were >5 cm in size, increasing in size in 44%, located deep to the fascia in 47%, and painful in 21%. These data contrast with those from specialist units where 39% of tumors were >5 cm, 70% were growing, 53% were deep to the fascia, and 32% were painful. Others have analyzed the correlation between data obtained in PC vs. specialist units, finding a high correlation (82%) in size estimation in STS > 5 cm. The mean STS size in PC was 7.6 ± 4.3 cm vs. 8.6 ± 5.9 cm in specialist units. Despite all this, it is surprising to find a very low complementarity of referral documents. In Japan, a study [113] of 142 STS patients found that tumor size, depth, and both were described in 51.4%, 36.6%, and 23.2% of referrals, respectively. The study also shows that orthopedic surgeons are more aware of the importance of reporting tumor size than non-orthopedic surgeons.

## 8. Referral Criteria

Several organizations have established different clinical criteria for early referral (Table 7), with size and deep location being the most consistently considered symptoms [24,97,107,108,114,115,116,117]. The use of a combination of symptoms as referral criteria carries the risk of creating criteria that are either too narrow or too broad. Alarm symptom selection should identify all sarcoma patients to ensure timely diagnosis and treatment, without including too many patients with benign conditions. The use of too narrow criteria may lead to exclusion of patients without the selected alarm symptoms, whereas too broad criteria may saturate the system with unnecessary referrals. GPs and referrers need to be aware of referral guidelines and have clear and sufficient criteria to select patients for referral.

The use of clinical criteria for referral has an important disadvantage for patients who do not present with alarm symptoms, as an alarm symptoms-only system will only benefit symptomatic patients. Dyrop et al. in 2014 [118] found that at least 35% of STS cases had none of the alarming symptoms. In this case, these patients are referred out of the urgent pathway, which means an unavoidable delay. The addition of other clinical findings, such as regrowth of a previously removed tumor or changes in a tumor that has been present for years, should also be considered. Another strategy is the addition of a referral guide for non-specific symptoms to capture patients with atypical presentations and the introduction of a guide for and yes-no clinics with easy access to imaging for GPs.

One of the most popular referral guidelines comes from NICE in UK [33,119], which initially identified “red flag” symptoms for soft tissue tumors, considered to be the “classic red flags” for referral, namely, a growing and painful mass > 5 cm in size and deep in location. This guidance for referral has been updated to reflect important changes such as the removal of location and pain and the inclusion of age (>50 years) and short duration of symptoms. In the latest update (October 2023) [120], the referral criteria have been further modified to recommend urgent and direct access ultrasound (within 2 weeks) to assess for STSs in adults with an unexplained lump that is increasing in size. It is also recommended that these patients be referred if they have ultrasound findings suggestive of STSs or if the ultrasound findings are equivocal and clinical concern remains. As we can see, we have gone from strict clinical criteria to much simpler criteria, based specifically on imaging data. In addition, the NICE makes some important considerations. Specifically, there are no sarcoma-specific signs/symptoms with a PPV > 3%, so only radiological criteria are included in the referral criteria; greater freedom is given to PC in the clinical assessment of patients; radiological referral criteria mean that many patients may be “lost” as GPs do not have easy access to radiology; imaging techniques should be performed by radiologists with experience in musculoskeletal radiology; and finally, and despite the above, never forget the “classic” warning signs.

## 9. Fast-Track Referral Pathways

There are several guidelines describing the pathway that patients must follow from PC (GPs), or where patients are assessed, to access the sarcoma MDT if required. One concern about the introduction of urgent and rapid pathways is that patients may react negatively to the accelerated diagnostic process, which leaves little time for reflection and adaptation. Although this problem is real, most patients prefer to be diagnosed as soon as possible, a factor that reduces their potential anticipatory anxiety.

One of the most commonly described referral guidelines is the “Two-Week Wait Guideline” (2WW), which was introduced in England and Wales by the Department of Health in 2000 [114,119,120]. The guideline states that at least 93% of the patients should be seen by an MDT within two weeks and that a diagnosis should be made within 31 days of referral. For those with proven malignancy, treatment should then begin within a further 31 days. This referral pathway is accompanied by the NICE guidelines for referral [119,120], which include recommendations on alarms symptoms qualifying for a 2WW referral. The key target is to see a specialist within 14 days of urgent referral. The target is not how long someone waits for tests, very frequently not in a timely manner; when the results are available; and, most importantly, whether they have cancer or not. The target is to see a specialist, not obtaining a diagnosis, meaning that the final diagnosis can take much longer than two weeks. Other drawbacks are the difficulties encountered to implement new tests, as available complex tests often take longer than two weeks to obtain results and return them to doctors. As a result, these tests are not used in favor of older tests that provide results more quickly. Finally, the 2WW target of 93% is not being met, as compliance rates in England are around 75%. First proposed in 2015 as part of the NHS Long Term Plan, the “Faster Diagnosis Standard” (FDS) [121] replaced in 2023 the 2WW referral as the FDS addresses these issues and in particular the main target, with the goal that 75% of all people referred via an urgent suspected cancer referral should receive a non-cancer or confirmed cancer diagnosis within 28 days. In addition, the FDS includes two other targets: 31-day decision to treat to first treatment target (this refers to the date the clinical team and the patient agree on the treatment plan to the date treatment starts) and 62-day referral to first treatment target (for those who receive a cancer diagnosis, the 62-day target starts from the date the referral is received to the date the person starts their cancer treatment).

Similar initiatives have been described in other European countries. In Spain, although not as extensive as the 2WW, it is possible to describe several guidelines [122,123,124], some of these including alarm symptoms as referral criteria, direct access to specialists, and a 30-day target from suspicion to treatment. Guidelines for referral in the Netherlands were also described [116,125], without time limits for referral. In Sweden, simple guidelines for sarcoma referral based on alarm symptoms have been used. An open-access outpatient clinic is available for GP referral, but no time limits are defined [108]. In Scotland, the “Scottish Sarcoma Managed Clinical Network” (SSMN) was established in 2004 to provide designated surgical and oncological centers to facilitate patient access without compromising quality of care [115]. They include warning symptoms but no defined time limits. Of particular interest is the CPP guideline implemented in Denmark in 2007–2008 (Figure 3) [24], which includes some of the same components as the 2WW, such as referral criteria for alarm symptoms and timeframes, including the goal of ensuring timely diagnosis, which is the same. However, the CPP includes more of the diagnostic process than the 2WW pathway. The main aims of the CPP are to increase cancer survival by reducing system delays in diagnosis, to improve patient satisfaction and health outcomes through prompt treatment, and to reduce patient distress caused by unnecessary waiting and ensure continuity of care. The CPP is organized in a different way because the GP may not refer patients directly to the CPP on the basis of clinical findings alone. The patient should be referred to a local orthopedic hospital department for clinical assessment and imaging (preferably an MRI and also an X-ray for bone tumors), and only if the imaging raises a suspicion can the patient be referred to a sarcoma center. The CPP starts when the referral is received at the sarcoma center, not when the symptoms are detected by the GP, at which point the time limits described in Figure 3 apply. This guideline has been quite successful because of the unique collaboration between politicians and health professionals and the fact that its implementation has been backed by a large financial investment. Ongoing compliance is ensured because waiting times are monitored by the government, and if the proportion of patients exceeding the time limits becomes too high, the responsible department can be sanctioned or have financial support withdrawn.

## 10. Improving and Promoting Early Referral—General Measures

In order to implement actions to promote early diagnosis and referral, it is essential to know where to focus efforts. In Part I of this review, we recognized that the patient and non-specialist delay was by far the longest. According to Elyes et al. [29], the patient interval is the longest of the total diagnostic time and is considered a “bottleneck”, followed by the secondary care interval, another well recognized “bottleneck”. Lack of awareness and knowledge of symptoms/signs in STSs, including lack of information about the existence of referral guidelines or the existence of an MDT, are the main barriers to early referral.

Several initiatives have been developed to improve knowledge and awareness of STSs. One of the most important initiatives was the “National Audit of Cancer Diagnosis in Primary Care”, developed between 2009 and 2010 [126,127,128], in which we can find several recommendations. In terms of general health education, it was recommended to promote regular health campaigns focused on increasing suspicion of sarcomas through the identification of “red flag” symptoms with encouragement to self-examine; the identification of “red flag” symptoms with higher PPV; and the inclusion of physical examination (visual + palpation) as mandatory in workplaces/schools. In PC, recommendations included the availability of complementary testing, PC assessment, the use of decision support tools/risk calculators, and the encouragement of PC consultation in DM/smokers with soft tissue tumors. STS education was another point of interest, with recommendations including the use of informative posters on symptoms/signs of STSs/bone sarcomas; the implementation of routine seminars and clinical sessions on STSs and dedicated sarcoma guideline courses with 1 month, 6 month and 1 year follow-up to promote knowledge retention; and increased education in the field of sarcoma at national congresses with continuous feedback. The following were also recommended: the creation and dissemination of guidelines and referral circuits; improved awareness of the existence of sarcoma units; the need for strong political support; and the creation and promotion of the figure of “clinical or case coordinator”, as an active figure in avoiding delays, but without clinical capacity, with a very positive impact economically and socially, avoiding unnecessary diagnostic procedures. Other recommendations that focus specifically on the health system are to facilitate better access to diagnostic tests in PC and the use of cancer risk scores, such as the QResearch database or the Risk Assessment Tool, to facilitate referral decisions. Clinical referral guidelines must to be easy to apply, without prior imaging or biopsy to avoid false negatives/false positives due to inexperienced surgeons/pathologists. Pain or tumor growth must also be excluded from the guidelines, but it is recommended to include the possibility of direct referral from PC to MDT (telephone). Finally, it is recommended that specialized centers improve diagnostic intervals; avoid loss of time in local hospitals and use appropriate imaging techniques.

According to “The Sarcoma Policy Checklist”, published in 2017 [129], there is a need for more professional training: development and dissemination of a national referral protocol for suspected sarcoma patients, advising non-specialists on “red flag” symptoms and when to refer patients to centers of reference; training on rare cancers in the general medical curriculum; ongoing training on rare cancers available to all oncologists; and specific training programs on sarcomas available to all health care professionals involved in the sarcoma multidisciplinary care team. Finally, the “Sarcoma UK 2020” report [1] displays a list of recommendations to improve education of health care professionals and public awareness on sarcomas and to create effective and efficient sarcoma referrals guidelines.

Because of the extraordinary medico-legal implications of diagnostic delay (see part I), several recommendations were made from a legal perspective [86,87,88,89,90,91,92,93,94,95]. GPs should carefully review all imaging studies to avoid missed suspicious findings and incidental findings in asymptomatic patients. Clinicians and health care institutions should establish communication protocols to clarify critical findings from diagnostic studies. It is also important to follow radiologists’ or pathologists’ recommendations for further testing. It is very useful to ask for a second opinion, especially in radiology and/or pathology. Finally, is it important to establish clinical guidelines for early referral.

## 11. What Are the Results After the Implementation of Referral Clinical Criteria and Fast-Track Referral Pathways?

Malik et al. [130] in 2007 analyzed 547 referrals to the North of England Sarcoma Service in Newcastle between 2004 and 2005. The authors found that only 7.8% of referrals and 15% of sarcomas finally diagnosed followed the 2WW pathway. In the same regard, Taylor et al. [131] in 2010 analyzed 2695 referrals in Birmingham. They found that 35% of referrals and 13% of STSs were present throughout the 2WW pathway. Pencavel et al. [132], from the Royal Marsden Hospital in London, analyzed 2746 referrals between 2004 and 2008. Again, 2WW referrals were only 5.6% of the total, and 2% of STSs were diagnosed throughout the 2WW pathway. Other findings of interest were that the number of total referrals increased up to 25-fold in 4 years with only a 0.3% increase in STS diagnoses. This is the “detection rate” (DR), i.e., the percentage of new cancer cases treated as a result of a referral. In addition, the number of benign soft tissue tumors investigated increased from 20% in 2004 to 30% in 2008. Finally, Szucs [133], from Cambridge University Hospital NHS FT, showed that only 1 in 10 tumors referred via the 2WW guide were STSs (only 13% of STSs were referred via 2WW), with a very large median size (10.6 cm). In conclusion, the authors felt that although it is possible to achieve a “conversion rate” (CR), i.e., the percentage of referrals that result in a cancer diagnosis, around 10%, the use of the 2WW pathway is rare and unable to achieve early diagnosis. The authors [130,131,132,133] considered these results to be hopeless, and it is possible to identify several drawbacks of this clinical referral guideline. First, adherence to the guideline is low, and long delays for sarcoma patients persisted after its implementation. Waiting times have increased because many sarcoma patients do not meet the referral criteria and are diagnosed outside the 2WW. Finally, the 2WW has not led to more sarcomas being diagnosed, only to an increase in patients referred with benign conditions, taking up valuable sarcoma specialist capacity with a clear risk of collapse [130,131,132,133].

A 2015 study by Gerrand et al. [134] analyses the “Route to Diagnosis” (RTD) that patients with suspected sarcoma follow in England in the 6 months before cancer diagnosis. This is a retrospective study using data from the National Cancer Data Repository from 1990 to 2010. This repository collects data from the eight English regional cancer registries. The key findings of the study were that 39% of patients presented via GPs referrals, 22.1% presented through 2WW, 16.1% were emergency presentations (EPs), and 15.5% presented through other outpatient appointments. These data confirm that sarcomas often go unrecognized in PC, and many patients are not referred even when referral guideline criteria are met. The high rate of inpatient elective RTD (7% in STSs vs. 2% for all cancers) suggests inpatient investigation is needed before sarcoma is considered in a proportion of patients. Finally, it was observed that there was an increase in patients diagnosed via the 2WW pathway: 22.1% compared to 12.5% in 2006–2008. Looking specifically at STSs, it was observed that patients under 10 years, those aged 10–19 years, or those with 80 years or older of age were most likely to have an EP RTD (54%, 28%, and 25%, respectively). Perhaps we are facing the consequence of particularly aggressive tumors that are very prone to developing multiple complications, forcing patients to go to emergency departments. In fact, these patients have a higher risk of metastatic disease (29% versus 9% in the case of GP referral). The authors concluded that there is an interaction among age, tumor type, anatomical location, and socioeconomical status that leads to different RTDs according to the combined characteristics of the patients. The results of a recent analysis of the National Cancer Registration and Analysis Service between 2013 and 2017 in England [3] allow similar conclusions to be drawn: 39.0% of patients presented via GP referral, 22.1% via 2WW, 16.1% via EP, and 15.5% via other outpatient appointments. Patients with rhabdomyosarcoma and GIST were most likely to present via EP (36.3% and 26.4%, respectively). In the same regard, children/AYA and the oldest cohort present via EPs, and an increasing likelihood for patients to present via EPs was noted as deprivation increases.

An issue of interest is patient non-attendance at referral appointments. In a study by Sheridan et al. [135] that analyzed 109,433 patients referred to a cancer center from 2009 to 2016, although not specifically for sarcoma, 5673 (5.2%) did not attend. Factors associated with non-attendance were younger and older age, male gender, greater deprivation, suspected cancer site, earlier year of referral, and greater distance to hospital. This issue has important prognostic implications, as patients who do not attend have an early mortality risk of 31.3% compared to 19.2% for those who do attend.

### Has the Implementation of These Guidelines Changed the Referral and Characteristics of STSs Seen by the Sarcoma MDT?

Smith et al. from Birmingham [136] studied the impact of the 2005 NICE guidelines between 1985 and 2009, analyzing 4934 soft tissue tumors, of which 48% were STSs. The authors observed a reduction in tumor size from 10.2 cm to 9.6 cm, with no change in the duration of symptoms (26 weeks) or the incidence of “whoops” resections (25%). The authors considered these results to be hopeless. Fujiwara et al. [137], also from Birmingham, analyzed the impact of the NICE guidelines by analyzing 2,427 referred STSs from 1996 to 2016. Comparing a pre-NICE group (1996–2006) with a post-NICE group (2006–2016), the authors observed a reduction in “whoops” resections from 28% to 20%, a reduction in median size (from 10.3 cm to 9.1 cm), and a reduction in the time between referral and first visit from 16.2 days to 15.4 days. There was also an improvement in 5-year survival (63% vs. 71%).

Several other groups have analyzed the impact of other referral guidelines in different health systems and countries. McCullough et al. [115] observed the impact of the implementation in 2004 of the SSMN in Scotland by analyzing 158 patients divided into two groups (before and after 2004). The authors observed, along with good adherence, a positive effect with a reduction in the delay from referral to first visit to the MDT from 19.5 days to 10 days, an increase in the use of MRI from 67% to 86%, an increase in preoperative biopsy from 57% to 79%, and a reduction in “whoops” procedures from 81% to 48%. In 2012, Styring et al. [108] analyzed 564 patients in Sweden, showing a good adherence with a progressive increase in the number of referred cases, with the majority (75%) being referred <3 months after the initial GP visit. It was also observed that 100% of deep STSs and 66% of superficial STSs were primary (no surgery or biopsy) and 33% of superficial STS <5 cm were referred after “whoops” resection. The authors analyzed the potential saturation of the system. However, they found that for every STS, there were 3 benign tumors, of which 33% were treated at the center. The authors concluded that the implementation of the guidelines were positive. Jansen-Landheer et al. [116], in 2005, analyzed the results in The Netherlands, observing low adherence rates that improved after its standardization. An increase in resections in specialized units (37% vs. 50%), a decrease in re-resections (24% vs. 18%), a reduction in suboptimal treatments (19% vs. 6%), an increase in the use of preoperative biopsies (9% vs. 53%), and a reduction in no pathological diagnosis (38% vs. 25%) were also observed. In Spain, reports on other cancers indicate that the time to diagnosis was reduced after implementation of guidelines [122,123,124]. The Spanish application SCAE-SM (appointment request for specialist care—malignant suspicion) [138] allows GPs to ask for a prompt appointment (less than 2 weeks) with a sarcoma specialist, but a recent analysis showed very poor results with 18.8% of sarcomas being referred through this way with a mean delay of 18 days. Another study reported long delays for sarcoma patients, despite the existence of referral guidelines. In 2013, Dyrop et al. [139] analyzed the impact of the implementation of the CPP guidelines in Denmark. Analyzing 1126 patients from 2007 to 2010, the authors observed a reduction in cases >5 cm from 22.6% in 2007 to 14.2% in 2010, and a reduction in waiting times, with on-time intervals (according to their recommended maximum delay times) increasing from 51.2% to 82.7%. The impact of these guidelines was finally reviewed by Thorn et al. in 2024 [25]. The authors compared results between two time periods (pre-CPP 2000–2008 vs. post-CPP 2010–2018), analyzing 2705 adult patients with deep-seated HG STSs in the extremities and trunk. Specifically, 5-year OS improved from 43% to 52%, and TTI was significantly reduced (median reduction: 3 days—18 vs. 15 days). In addition, the number of unplanned resections was also reduced (10% vs. 4%), and the proportion of patients receiving RT and CT increased from 47% to 57% and from 28% to 41%, respectively. In contrast to what was reported by Dyrop et al. in 2013 [139], no reduction in tumor size was observed, a finding that the authors linked to the possibility of reaching a plateau.

## 12. Avoiding Collapse—How to Filter Massive Referrals

Clinical referral guidelines and pathways for patients are well established, being clearly described the signs/symptoms that can alert us that a lump may be an STS. All the information campaigns and recommendations already described have as their main aim to improve referral rates throughout the early suspicion of STS, but they can have several counterproductive and undesirable effects. First, the saturation of MDTs due to the overwhelming number of referrals. It is calculated that the number of referrals for suspected cancer increased by 26% between 1997 and 2007 [3,8]. Using the latest reported data [3,8], there are more than 2.7 million referrals for suspected cancer each year in England with an increase for sarcoma referrals from 6.7 per 100,000 in 2009/2010 to 21.8 per 100,000 in 2019/2020. However, these positive developments come at a cost: a decrease in CR from 11.3% to 6.5% over the same periods (only around one in 15 referred patients being found to have cancer) and an increase in DR from 25.9% to 44.8% [3,8]. Second, it appears that information campaigns have had an unexpected effect. GPs are aware of the seriousness of the disease and the role they play in reducing the time to diagnosis. However, at the same time, due to the low incidence and prognostic severity of STSs, they are not confident in its management or in applying the information received, preferring to refer the patient rather than to do so according to the established criteria. This situation can be aggravate because almost half of symptomatic patients who go on to be diagnosed with cancer do not have “red flag” symptoms, guidelines may be exceeded when clinicians consider patients to be at risk of undiagnosed cancer, and, finally, non-adherence to referral recommendations is common. All of this means that a full-time GP will be making around 65 referrals per year to detect 4–5 cancers (7% CR), putting diagnostic services, i.e., radiology, under a lot of pressure. Third, as the number of referrals increases, the number of diagnosed STSs decreases significantly [3,8].

In this situation, it is important to manage referrals effectively to ensure that only appropriate referrals are discussed by the MDT to keep DR as high as possible with CR in reasonable levels. Neal [140] suggests five different areas to achieve this. First, it is important to ensure compliance with existing guidelines, which can be improved through the use of tools such as “C The Signs”, which enables practices to benchmark and audit their referral performance, and interventions such as “ThinkCancer”, which aims to get GPs and practice staff to consider the possibility of cancer more often. Second, it is imperative to develop pathways for low-risk symptomatic patients who do not qualify for urgent 2WW referral “low-risk but not no-risk patients”. Third, it is important to monitor site-specific urgent referral pathways. Fourth, there is a need to expand direct access to GPs for diagnostics, particularly imaging. Fifth, it is important to innovate in biomarkers and artificial intelligence/machine learning to identify and prioritize patients at highest risk, as outlined in the Roadmap for Early Detection and Diagnosis of Cancer.

### 12.1. Pre-Referral Imaging

The most commonly used strategy is to use pre-referral imaging to select patients with a high likelihood of sarcoma for MDT discussion. These images can be assessed according to well-established pre-referral criteria, by incorporating imaging into clinical referral guidelines; by reviewing these images in diagnostic triage, i.e., the pre-assessment of patients by a specialist before they attend the MDT; and by the use of telemedicine (Figure 4).

Ultrasound is the primary screening tool for the evaluation of soft tissue lesions, with MRI generally reserved for the characterization of indeterminate or suspicious lesions [141]. A study from Leeds in 2009 [142] confirmed that with its use, up to 79% of patients can be classified as benign. MRI is considered to be the ideal technique in the case of deep lesions that are inaccessible by ultrasound or where ultrasound has shown indeterminate/suspicious findings. In this case, it is recommended that MRI be performed at a specialist center, not at local hospital using ultrasound, for better characterization [141].

Several authors [131,142,143,144,145] have shown excellent results with the use of pre-referral imaging (Figure 5). A 2010 study in London [132] showed a 2% CR when referrals were based on clinical data alone, compared to 17% with CT/MRI. In Birmingham [131], the results were similar, with a 13% CR when referrals were based on clinical data alone, compared to 49% with CT/MRI. Rowbotham et al., in Leeds (UK) in 2012 [143], analyzed 112 soft tissue tumors, defining the delay in image referral as 10 days. The authors found that 18% exceeded this limit with ultrasound, 39% with MRI, and 43% with ultrasound and MRI. The overall average delay was 11.7 days, with 67% of patients having no delay. Other results of interest were a reduction in benign lesions (from 95% to 85%), a reduction in cystic lesions (from 34% to 5%), an increase in biopsy rate (from 8% to 44%), and an increase in CR (from 2% to 15%). The authors concluded that without supportive radiology there would be unnecessary patient anxiety and the need to attend the MDT, reducing the ability of the MDT to manage suspicious/indeterminate lesions. Finally, the authors recommend that the imaging request should remain with the PC to further reduce delays. In Denmark, Dyrop et al. in 2017 [144] analyzed 545 patients and showed an increase in delay time from 64–93 days to 166–189 (+91 days if CT/MRI is performed), an increase in CR from 7% to 17–24%, and a low repeat test rate of 3.5%. Finally, Ballhause et al. in 2022 [145] analyzed 302 soft tissue tumors in Germany, of which 86% had undergone MRI outside the specialist center. The authors show a concordance between MRI and final pathology of 67.5%, with a delay of 31.3 days for benign findings, 24.1 days for intermediate findings, and 14.8 days for malignant findings. The authors concluded that radiologists need to recommend referral or biopsy and that the use of telemedicine needs to be encouraged.

Despite these results, not all countries have adopted this strategy because of two main problems: poor quality and interpretation of imaging performed outside the specialist center, which sometimes requires repeat radiological tests, and delays at the local hospital. Both problems clearly linked to the organizational structure of the health care system. In Denmark [144], the strategy of imaging in local hospitals was found to increase the median diagnostic interval by 41 days and the total median interval by 91 days. The delay associated with the time spent in local hospital imaging can be addressed by changing the way imaging tests are performed, including shifting from serial to parallel investigations, where all investigations are started at the same time. Other options may be to include local hospital investigations in the fast-track program, which sets recommended and reasonable time limits, or to give GPs easier access to imaging investigations.

### 12.2. Diagnostic Triage

The East Midland Sarcoma Service developed a strategy that resulted in the creation of the “Sarcoma Diagnostic Triage Meeting” (SDTM) in 2010 [146]. The SDTM comprises a radiologist, an orthopedic surgeon, a pathologist, and a case coordinator. The SDTM acts as a filter, reviewing available prior imaging (usually ultrasound) on all new referrals and classifying them as benign, indeterminate, or suspicious. If benign, the patient is removed from the referral pathway and discharged, or, if necessary, a routine follow-up appointment is arranged. If indeterminate, a repeat ultrasound is performed by a musculoskeletal radiologist. Suspicious lesions will be referred for urgent MRI, if not already done, and urgent biopsy, if appropriate. For both indeterminate and suspicious referrals, the patient is re-discussed in the SDTM after complete imaging with or without histological review. Only biopsy-proven STSs were referred to the MDT. Shah et al. [146] published the results of their application in 2015, showing that after analyzing 165 initially referred cases, 87% were not ultimately reviewed by the MDT, at a cost saving of approximately GBP 800,000. In the same regard, Alanie et al. [147] analyzed the results of the implementation of the “Virtual Sarcoma Referral Model” in the West of Scotland Sarcoma Service in 2010. In this case, the meeting involves at least one consultant orthopedic surgical oncologist, one consultant musculoskeletal radiologist, and one orthopedic oncology clinical nurse specialist. The authors analyzed 5714 events (3976 patients) between 2010 and 2018 and showed that despite an increase in referral volume, only 26% of events discussed were ultimately discussed at the MDT, with a potential cost saving of GBP 74,522 per year. The authors considered this to be a cost-effective way of reducing unnecessary referrals.

### 12.3. Telemedicine

Despite the fact that benign tumors may mimic malignant counterparts, it is possible to identify radiological criteria suggestive of STS, such as inhomogeneous density due to hemorrhage, necrosis, walls, and calcifications. Based in these findings, the French Sarcoma Group (FSG) team of radiologists developed an algorithm to differentiate between benign and non-benign tumors based on imaging [148]. In 2021, a mobile app (“Sar’Connect”) was developed to increase the rate of early detection of STSs and facilitate patient referral based on the FSG radiological algorithm and the geolocation of health care professionals. The FSG radiological algorithm consists of a list of 3–6 short answer questions based on clinical and radiological information. After completing the list, the user receives a recommendation: direct referral, supplemental imaging, and possible non-expert center management. In 2022, Nannini et al. published the first results of this strategy [148], comparing Sar’Connect advice with real-life management. A difference of 51.7% was observed in the management recommendation, with a gain in time to referral of 7.7 months using the application (6.5 months for non-benign tumors) and a better guidance for benign tumors of 22%. The authors concluded that this strategy can help to reduce time to referral of potentially malignant tumors and avoid collapse of the MDT through better management of benign tumors.

This is an example of how telehealth/digital health tools can help improve referrals to the MDT [149,150]. The use of this approach has increased greatly since the COVID pandemic because of its benefits in the field of sarcoma, allowing easy and rapid communication between physicians and MDT, reducing delays in follow-up appointments, mitigating the effects of living a long distance from the referral center, and reducing the effects of low socio-economic status. A recent systematic review on telemedicine in sarcoma care concludes that telehealth offers opportunities for tailored and individualized care playing a crucial role in sarcoma management.

## 13. Conclusions

Diagnostic and therapeutic delay in STSs is a very common problem with important implications, from prognostic to legal or psychosocial. After analyzing the causes, the common denominator is the lack of knowledge and awareness. The presence of health system barriers often complicates the situation. We believe it is essential for every sarcoma team to identify the extent of the problem and its causes. Without this information, it is almost impossible to properly address the problem and take corrective actions. The prevention and management of this problem should be based on the dissemination of knowledge about STS and its key warning symptoms/signs, raising awareness of STS achieved by public information campaigns, and the inclusion of STS in ongoing education of medical students and specialists in training. The implementation and dissemination of referral guidelines and fast-track pathways can help to improve early referral. To reduce the risk of MDT collapse due to over-referral, the use of pre-referral imaging or diagnostic triage meetings as a filter for access to the MDT has been shown to be an excellent strategy. Other actions of interest include the need for more research into the presentation and prevalence of musculoskeletal tumors in PC as well as the need for continued vigilance regarding time limits and adherence to referral guidelines. All these processes can be improved by better use of referral centers (better definition, training in PC, and promotion of sarcoma awareness) and decentralization of care to those who do not travel (centralized pathology review and centralized surgery only for the most appropriate cases).

## Figures and Tables

**Figure 1 cancers-17-01861-f001:**
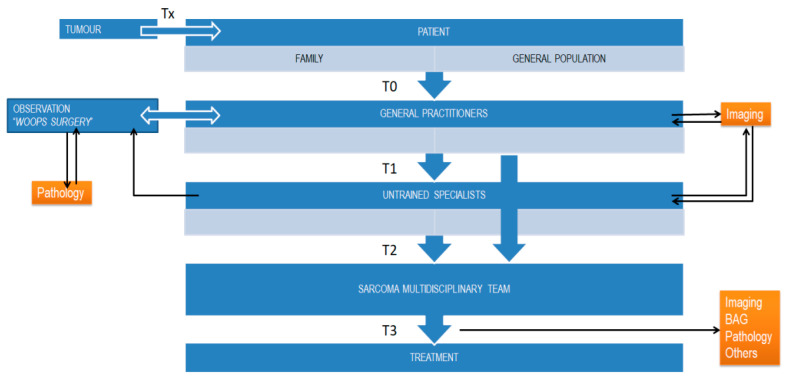
Simplified and schematic sarcoma model in which we can identify the main actors involved in the diagnostic process, the time intervals considered, and the temporal distribution of the delay.

**Figure 2 cancers-17-01861-f002:**
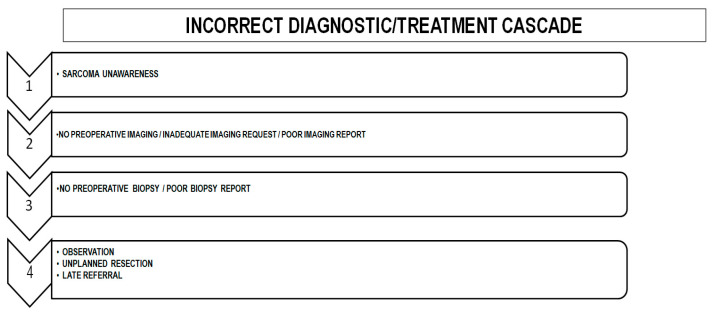
The fatal diagnostic/treatment cascade in sarcomas.

**Figure 3 cancers-17-01861-f003:**
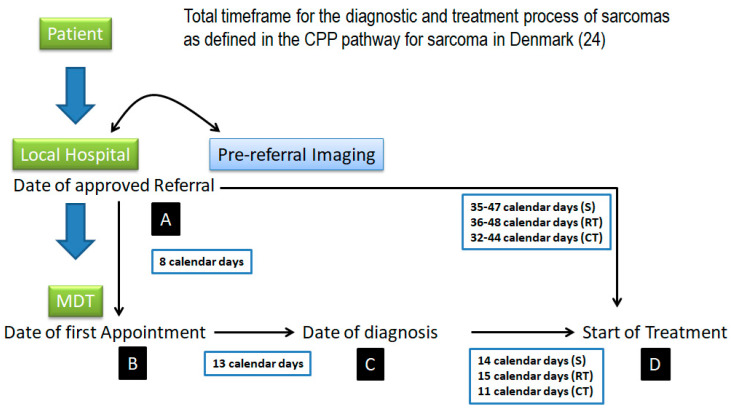
Danish Cancer Patient Pathway and recommended lapse times. A–B, Time lapse between local hospitals to sarcoma team; B–C, time lapse between first consultation with sarcoma team and final diagnosis; C–D, time lapse between diagnosis and start of treatment.

**Figure 4 cancers-17-01861-f004:**
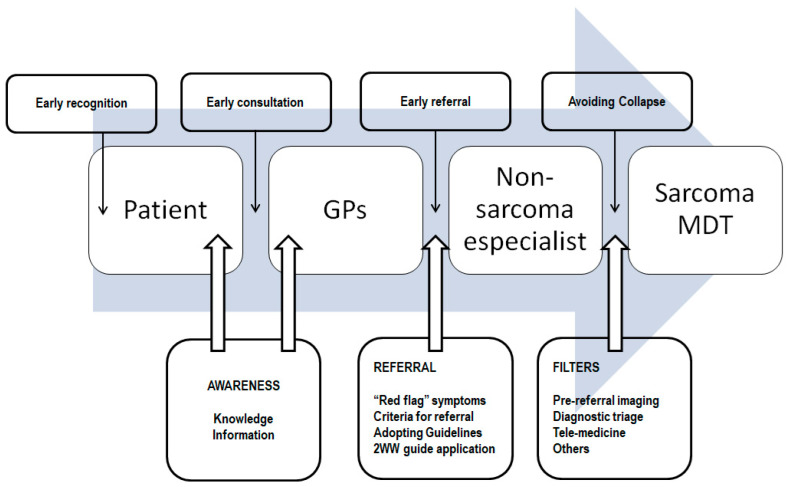
Cancer Patient Pathway and main measures to improve referral avoiding collapse.

**Figure 5 cancers-17-01861-f005:**
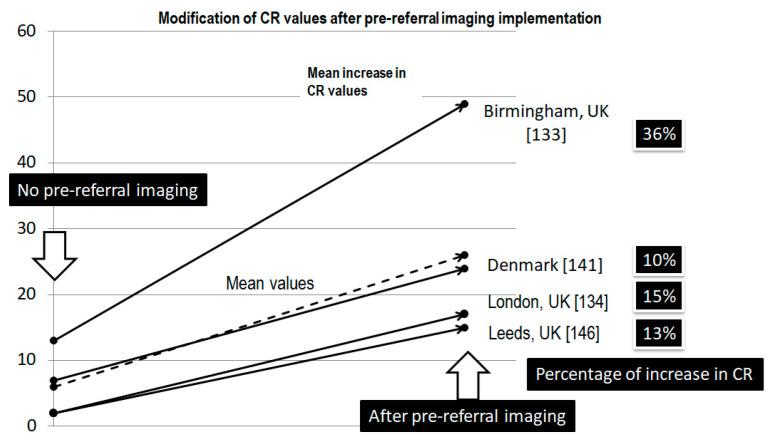
Main results observed with the use of pre-referral imaging.

**Table 1 cancers-17-01861-t001:** Aarhus Statement from the Consensus Working Group, 2009 [20]. Commissioned Cancer Research UK and the Department of Health in England under the auspices of the Cancer and Primary Care Research International (Ca-PRI) Network.

Date of first symptom: the time point when first bodily changes and/or symptoms are noticed
The term “patient delay” should no longer be used: instead, “appraisal interval” (time taken to interpret bodily changes/symptoms) and “help-seeking interval” (time taken to act upon those interpretations and seek help) are more helpful in describing the “patient interval”
Date of first presentation: the time point at which, given the presenting signs, symptoms, history, and other risk factors, the patient has started an investigation or referral for possible important pathology, including cancer
Date of referral: the time point at which there is a transfer of responsibility from one health care provider to another (typically, in ‘gatekeeper’ health care systems, from a primary care provider to a doctor/service specializing in cancer diagnosis and management) for further clinical diagnostic and management activity, relating to the patient’s suspected cancer
Date of diagnosis: Any time interval that either begins or ends with “diagnosis”

**Table 2 cancers-17-01861-t002:** (**a**) Diagnostic delay in sarcoma: multi-institutional series. (**b**) Diagnostic delay in sarcomas: Single-institution series.

(a)								
Author, Year [Reference]	Database, Country	Years of Analysis, n	Type of Sarcoma	Patient Interval (T0)	Primary Care Interval	Secondary Care Interval	Tertiary Care Interval	Total Interval
Carrillo- García, 2024 [28]	Multi institutional (5), Spain	2018 (7 m) n = 182	BS: 19.2% STS: 80.8%	From symptoms to diagnosis: 83 days (5–328) From symptoms to surgery: 54.5 days (2–331)				
Elyes, 2023 [29]	Swiss Sarcoma Board (MDT/SSN), Switzerland	2018–2021 n = 1028	STS and BS benign tumors	8.8 w	0.4 w (0–1.4)	4.3 w (2.1–9.1)	1.3 w (−0.6–3.4)	23.3 w (10.4–59.4)
Soomers, 2020 [30] Cas Drabbe, 2021 [31]	Netherland Cancer Registry (NCR)/SURVSARC Study, The Netherlands	2008–2016 n = 1099	STS: 76% BS: 24%	>1 m: 60% >3 m: 36% >1 y: 15%	<2 w: 51% >2 w: 49%	<1 m: 64% 1–3 m: 23%	<1 m: 85% 1–2: 30%	Diagnostic interval: ≥1 m: 55% ≥3 m: 28% ≥1 y: 5%
Soomers, 2020 [32]	Systematic review of PubMed, Medline, Embase, Web of Science, and Cochrane Library	From 1945 36 studies n = 16,845	STS and BS	-	0.1–13.3 w	1.1–6.9 w	2.1–7.9 w	4.3–614.9 w
Sarcoma UK, 2020 [33]	Anonymous online survey via UK’s Sarcoma Networks	2020 n = 117	STS: 79% BS: 16% GIST: 5%	<2 w: 33% 2–4 w: 20% 4 w–3 m: 22% >3 m: 25%	-	-	-	Diagnostic interval: <2 w: 8% >3 m: 47%
Lyratzopoulos, 2015 [34]	English National Audit of Cancer Diagnosis in PC, UK	2009–2010 n = 72	STS & BS	45 days	45 days Pre-referral interval: 90 days			
**(b)**								
**Author, Year [Reference]**	**Centre, Country**	**Years of Analysis, n**	**Type** **of ** **Sarcoma**	**Patient Interval**	**Primary Care Interval**	**Secondary Care Interval**	**Tertiary Care Interval**	**Total ** **Interval**
Sam Martin, 2023 [35]	Sarcoma Unit, Royal National Orthopaedic Hospital, Stanmore, UK	2017 (5 m) n = 78	BS: 21 STS: 41 H/N: 9 GIST: 7	14 days (1–4971)				Diagnostic interval: 86 days (1–1276)
Dyrop, 2016 [36]	Aarhus Sarcoma Centre, Aarhus, Denmark	2014–2015 n = 102	STS	77 days (11–261)	17 days (1–56)	29 days (15–56)	17 days (10–24)	Diagnostic interval: 65 days (42–133) Total interval: 176 days (883–673)
Ramos- Pascua, 2014 [18]	Servicio de Cirugía Ortopédica y Traumatología, León, Spain	2006–2012 n = 112	Primary STS T/E	289 days	80 days	173 days	46 days	-
Nakamura, 2011 [37]	Department of Orthopaedic Surgery, Mie University Graduate School, Japan	2001–2009 n = 100	Primary STS No WD_LPS No DFS	3 m (1–72)	-	-	-	From symptoms to diagnosis: 6 m (1–72) TTI: 2 w
Seinen, 2010 [38]	Department of Oncology, Skäne University Hospital, Sweden	2003–2009 n = 33	Primary RPS LPS (40%) LMS (24%)	23 days (0–524)	15 days (0–244)	36 days (2–1223)	55 days (1–483)	-
Johnson, 2008 [39]	Royal Orthopaedic Hospital, Birmingham, UK	2005 n = 162	STS	1.3 w	2.4 w	9.1 w	-	Median time from symptoms to referral to sarcoma specialist: 40.4 w (mean: 2.3 w)
Clark, 2005 [26]	Soft-Tissue Sarcoma Unit of the Royal Marsden Hospital, London, UK	2003–2004 n = 159	STS and BS	-	14 m (4–96)	Hospital/specialist level delay 21 m		-
Brouns, 2003 [27]	Department of Surgical Oncology, University Hospital Gasthuisberg, Leuven, Belgium	1999–2001 n = 100	STS and GIST	4 m (1–240)		HG STS: 85% < 6 m, 50% w/o delay LG STS: 45% > 6 m, 50% w/o delay From first consultation to diagnosis: 6 m (2–79)		

STS, Soft tissue sarcoma; BS, bone sarcoma; GIST, gastrointestinal stromal tumor; WD_LPS, well differentiated liposarcoma; DFS, dermatofibrosarcoma; T/E, trunk/extremities; RPS, retroperitoneal sarcoma; LMS, leiomyosarcoma; H/N, head/neck; m, months; w, weeks; y, years; w/o, without.

**Table 3 cancers-17-01861-t003:** Series reporting time to treatment initiation (TTI) values in sarcomas.

Author, Year (Reference)	Database, Country	Years of Analysis, n	Type of Sarcoma	TTI	TTI Increase
Schärer, 2023 [22]	Swiss sarcoma Network (SSN/MDT/SB), Switzerland	2018–2022 n = 266	S/D_STS (80.1%) BS (19.9%)	S_STS: 42 days (27–71) D_STS: 28 days (18–48)	-
Ryu, 2023 [40]	New Zealand Sarcoma Multidisciplinary Meeting (NZ-MDM), New Zealand	2011–2020 n = 223	STS	Median: 33 days	Median TTI 2011–2015: 27 days (11.5–37) Median TTI 2016–2020: 35 days (21–47)
Ogura, 2021 [41]	National Cancer Database (NCDB), US	2004–2015 n = 23,786	HG-STS T/E	Median: 14 days (0–32) Mean: 20.9 ± 27.5 days	Median TTI in 2004: 11 days Median TTI in 2015: 17 days
Curtis, 2018 [42]	National Cancer Database (NCDB), US	2004–2013 n = 41,529	STS	Median: 22 days Mean: 29.7 days	Median TTI in 2004: 20 days Median TTI in 2013: 26 days

MDT, Multidisciplinary team; S/D, superficial/deep; STS, soft tissue tumor; HG, high grade; T/E, trunk/extremities; BS, bone sarcoma.

**Table 4 cancers-17-01861-t004:** Causes of delay according to the actor involved.

Tumor-related causes	Malignant vs. benign tumors
	Anatomic location
	Deep
	Grade
	Histology
	Stage
Patient-related causes	Patient’s characteristics DM, tobacco Mental condition Age
	Socio-economic status Private vs. others Income status Housing
	Geographical barriers Time or distance from reference center
	Socio-cultural barriers Race Educational status Immigrants
General population-related causes	Lack of awareness
Primary care-related causes	Lack of initial suspicion Misinterpretation of imaging/biopsy
Health system-related causes	Lack of communication/relationships Lack of clinical guidelines Lack of continuity of care
Sarcoma team-related causes	

**Table 5 cancers-17-01861-t005:** Diagnostic delay in sarcomas and its prognostic impact.

Author, Year, [Reference]	Period of Time, n	Histology	Variable Measured	Results	Waiting Time Paradox
Ryu, 2023 [40]	2011–2020 n = 223	HG T/Extr No WD_LPS No SC-STS	TTI	TTI > 30 days DMFS rate 63% vs. TTI < 30 days DMFS rate 50% (*p* = 0.03) LRFS/DSS: ns	Yes
Ogura, 2021 [41]	2004–2015 n = 23,786 (NCDB)	HG T/Extr	TTI	TTI: 0–30 days, OS: 57.7%/TTI: 31–60 days, OS: 53.9%, HR: 1.08 TTI: 61–90 days, OS: 50.6%, HR: 1.11/TTI > 91 days, OS: 49.9%, HR: 1.22 (*p* < 0.05)	No
Araki, 2020 [72]	210–2017 n = 153	STS T/Extr No LG	T0, T1/2	T0 < 5 months 5-years MFS 95% vs. 49% (>5 months), *p* = 0.02 T1/2 ≤ 29 days 5-years MFS 95% vs. 49% (>29 days), *p* = 0.01	No No
Featherall, 2019 [73]	2004–2012 n = 8648 (NCDB)	HG localized STS	TTI	Linear association with OS Minimal HR (0.64): 42 days	No
Nandra, 2015 [45]	1985–2010 n = 4945	STS: 46%	T0	LG: 44 weeks (mortality: 2.3%, HR: 8.52) vs. 20 weeks (HG) (mortality: 17%), *p* < 0.0001	Yes
Urakawa, 2015 [74]	2001–2011 n = 152	No DFS No WD_LPS	T0+T1+T2	5-year OS (≥6 months): 95% vs. 66% (<6 months), *p* = 0.016	Yes
Rougraff, 2012 [75]	1992–2007 n = 381	STS T/Extr	Symptoms- treatment	No relationship with size, OS, or metastases	Yes
Nakamura, 2011 [37]	2001–2009 n = 100	No WD_LPS	Symptoms- diagnosis	5-year OS (<6 months): 77% vs. 59.7% (>6 months) 5-year DMR (<6 months): 76.5% vs. 38.8% (>6 months)	No Yes
Rougraff, 2007 [71]	1992–2003 n = 624	STS/BS T/Extr	Symptoms- diagnosis	HG: 17 months vs. 29.4 months (LG). No relationship with Size, OS, Location or Metastases Q4: 5-year OS 84% vs. 65% (HR: 0.6), *p* < 0.05 Q4: 5-year DFS 81% vs. 60% (HR: 0.5), *p* < 0.05	Yes No No
Saithna, 2007 [76]	25 years period n = 1349	STS	Symptoms- diagnosis S/SD	Additional week of symptoms improves the monthly OS by 0.2% S/SD ratio impacts OS with a HR: 1.4 (*p* < 0.0001)	Yes No
Gofman, 2007 [77]	1991–2004 n = 73	SS T/Extr	Symptoms- diagnosis	Delayed ≤1 year was superior to a delayed >1 years in terms of systemic control (HR:0.33), *p* = 0.037. OS was not affected	No
Ruka, 1988 [78]	1950–1984 n = 267	HG STS No visceral No DFS, KS No desmoids	Symptoms- treatment S/SD	Duration of symptoms without relationship with OS 5-year OS in S/SD ratio ≤ 1: 41% (49 months) vs. 28% (21 months) in S/SD > 1, *p* = 0.02 5-year DFS in S/SD ratio ≤ 1: 45% (40 months) vs. 31% (8 months) in S/SD > 1, *p* = 0.019 5-year DM in S/SD ratio ≤ 1: 28% (17 months) vs. 44% (48 months) in S/SD > 1, *p* = 0.0034	Yes Yes Yes
Soomers, 2020 [32]	1945–present 36 studies n = 16,845	STS and BS	Total interval (10 retrospective studies)	No effect on survival: 5 studies Improved survival rate with longer total interval (26 vs. 20 weeks): 1 study (STS) Improved survival rates with shorter total interval: 3 studies	Yes Yes No

NCDB, National Cancer Database; HG, high grade; LG, low grade; SC, subcutaneous; WD_LPS; well differentiated liposarcoma; BS, bone sarcoma; SS, synovial sarcoma; KS, Kaposi’s sarcoma; STS, soft tissue sarcoma; T/Extr, trunk/extremities; DFS, dermatofibrosarcoma; TTI, time to treatment initiation (from diagnosis to initiation of treatment); T0, patient’s delay; T1, general practitioner’s delay; T2, non-specialist’s delay; S/SD, size/symptoms duration ratio; DMFS, distant metastasis-free survival; LRFS, local recurrence-free survival; DSS, disease-specific survival; OS, overall survival; MFS, metastasis-free survival; DMR, distant metastasis rate; DFS, disease-free survival; DM, distant metastasis.

**Table 6 cancers-17-01861-t006:** Medico-legal impact of diagnostic delay in sarcomas.

Author, Year [Reference]	Database, Country, Years Reviewed, Number of Claims	Type of Sarcoma	Plaintiff (%)	Alleged Negligence	Negligence Proven	Average Indemnity Payment
Mesko, 2014 [89]	LexisNexis^®^, USA 1980–2012 n = 216	Extremity STS (59.7%) BS (40.3%)	PC (34.4%) OpS (22.8%) Rx (12.4%)	Delay in diagnosis (81%) Unnecessary amputation (10.6%)	62%	USD 2,302,483 (65,076– 12,661,611)
Ross, 2015 [91]	NHSLA, UK (England and Wales) 2003–2012 n = 69	NA	OpS (100%)	Delay in diagnosis (23%)	36.2%	GBP 116,502
Harrison, 2016 [90]	NHSLA, UK (England and Wales) 1995–2010 n = 52	Orthopedic related	OpS (100%)	Delay in diagnosis (66%)	71%	GBP 84,000
Hwang, 2020 [92]	Westlaw MDB (USA) 1982–2018 n = 92	All STS (57%) BS (43%)	PC (26%) OpS (23%) Rx (15%)	Delay in diagnosis (91%)	30% Settlement:32%	USD 1.9 millions
Davis, 2023 [93]	Westlaw MDB (USA) 1980–2021 n = 36	-	OpS	Failure diagnosis: 42%	Defense: 75% Plaintiff: 17% Settlement: 3%	USD 1,672,500 (134,231– 6,250,000)
Bayram, 2024 [94]	UYAP, TRK 2014–2024 n = 70	Primary STS/BS	OpS (51.7%) Pth (16.4%) Rx (8.2%)	Delay diagnosis and/or diagnostic error: 29.4%	11 cases (16%)	-

NHSLA, National Health Service Litigation Authority; UYAP, National Judicial Network Information System, Turkey; OpS, orthopedic surgeons; PC, primary care; Path, pathology; Rx, radiology.

**Table 7 cancers-17-01861-t007:** Clinical criteria for early referral according to main institutions.

	UK NICE 2005 [107]	UK NICE 2015 [114]	Sweden 2012 [108]	Scotland SSMCN 2004 [115]	The Netherlands 2005 [116]	Denmark CPP 2009 [24]	UK RNOH 2024 [9]	Japan JOP 2020 [117]
Size	>5 cm	>4.3 cm “Golf ball”	>5 cm	>5 cm	>3 cm and complex	>5 cm	>5 cm “Golf ball”	>5 cm
Location	Deep	-	Deep	Deep	-	Deep	Deep	Deep
Growth	Growing	Growing	-	Growing	-	Growing	Increasing in size	-
Pain	Painful lump	-	-	-	-	-	With or without concern in large painless lumps	-
Age	-	>50 years	-	-	-	-	-	-
Duration	-	Short	-	-	-	Short	-	-
Other criteria	-	-	No open biopsy or surgery No imaging	-	Radical excision in case of <3 cm Biopsy in deep >3 cm	Palpable bone tumor Deep and persistent bone pain	Firmer than surrounding tissues Risk factors *	-

* Radiotherapy, chronic lymphedema, and inherited syndrome such as Gardner syndrome, Li–Fraumeni syndrome, and Von Recklinghausen disease; more features indicate more risk. SSMCN, Scottish Sarcoma Managed Clinical Network; CPP, Cancer Patient Pathway; RNOH, Royal National Orthopaedic Hospital, Stanmore UK; JOP, Japan Orthopaedic Association Clinical Practice Guidelines.

## Data Availability

All data are available in the tables already displayed in this paper.

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
