# Peer review of "Diagnostic Delay in Soft Tissue Sarcomas: A Review"

_cancers, 2025, doi:10.3390/cancers17111861_

Round 1

Reviewer 1 Report

Comments and Suggestions for Authors

A comprehensive and thoughtful review. Some of the language is non-standard and would benefit from some editing from someone whose primary language is English.

Comments on the Quality of English Language

Needs editing for language.  Some of the sentences are unclear.

Author Response

We have reviewed and edited completely all the text to improve language

Reviewer 2 Report

Comments and Suggestions for Authors

The article  Diagnostic delay in STS A review is a complete analysis  of the reasons that can cause  most in HG  sarcomas  severe consequences in morbility and mortality.

The number of cited articles  is huge  and take in consideration  40 years of  publications. Congratulations.

However in my opinion there are  some weak points that the Authors  must consider.

1) The paper  is too long for a scientifical  Journal.  Forty one pages    of text can discourage many readers.

How to reduce the length?

a) Theoretical framework in diagnostic delay: it  is sufficient to cite the different models considered  without to   describe them scrupulously. In the   papers  of RCT  for instance the Authors  cite  the statistical model   considered ( Kaplan Meyer, Markov etc ) without  the need to describe   the model in detail. 

b) The reasons  of delay  should  be considered  resuming  in a table  the different causes: related to the tumor, to the patients, to the health organization etc . The text is too  full of data and difficult to  be scrutinized

c) The same problem  resurfaces in  the  extremely long chapter on the consequences of delay.

d) The problem of the  Referral Centers is well known: high quality of diagnosis and treatment but  long waiting list. In most cases Referral Centers are a heavy cause of delay themselves.

e) I agree that  telemedicine and virtual discussion can  accelerate the evaluation  of the case .Unfortunately  the real "bottle neck"  is not the multidisciplinary meeting  but the low number of expertised surgeons in STS and time to surgery.

g) Last point:  I suggest to exclude from the overview the retroperitoneal sarcomas. In this particular form of STS the delay is  determined  by the impossibility of an early diagnosis. In the STRASS study  retroperitoneal sarcomas were  usually diagnosed with large volumes,  the mean size identified was 16 cm.  and only the appareance  of abdominal symptoms  lead to suspect the  mass.

Author Response

ACCORDING TO THE REVIEWER #2, WE HAVE SHORTENED THE LEGHT OF THE TEXT AND, AT THE SAME TIME, CLARIFIED ITS COMPRENHENSION  BY SHORTENING THE THEORETICAL FRAMEWORK DISCUSSION ERASING THE TABLE; ADDING A NEW TABLE TO CLARIFY THE TEXT ABOUT CASUES OF DELAY, AND, FINALLY SHORTENING TEXT IN ALL ITEMS.  

Reviewer 3 Report

Comments and Suggestions for Authors

This review provides a timely and well-structured synthesis of diagnostic delay in soft tissue sarcoma (STS), integrating key interval models and proposing a multi-level blueprint for system improvement. It is grounded in real-world data and international literature, with strong clinical relevance and practical recommendations.

Major Points

1. The proposed system improvements (e.g., pre-referral imaging, SDTM, tele-triage) are valuable but currently scattered. Recommend summarizing them in a standalone visual or table for easier adoption.

2. While acknowledging inconsistent interval definitions, no attempt is made to harmonize them. A proposed standard definition table would enhance future comparability.

3. The review focuses heavily on UK/European systems. Suggest explicitly noting limitations for global transferability and contextual differences.

4. The 2025 Swiss SSN study on symptom-specific diagnostic delay (Elyes et al., Cancers 17:510) is highly relevant as it provides a multivariate modelling of how symptoms affect each diagnostic interval and may cited to strengthen the symptom-delay analysis.

Minor Points

  • Ensure clarity and readability of tables/figures, especially those illustrating pathway efficiency.

Accept with minor revisions – Implementation-oriented, insightful, and well-conceived; minor enhancements will further strengthen impact.

Author Response

According to reviewer #3, we have modified the text.

We have included new tables to summarise and clarify the text.
Table 1 and Figure 1 depict the most recent and widely accepted definitions of time intervals and the actors involved in the diagnostic process. We believe that adopting these definitions is essential if we want to compare results between studies, but it is also important to be able to modify them according to the needs of each sarcoma team.

According to reviewer #3, we have modified the text:

We have included new tables to summarise and clarify the text.

Table 1 and figure 1 depict the most recent and widely accepted definitions of time intervals and the actors involved in the diagnostic process. We beliver that adopting these definitions is essential if we want to compare results between studies, but it is also important to be able to modify them according to the needs of each sarcoma team. 

We have emphasized that most if not all literature is american or english which implies several limitations in terms of applicability in other countries.

We have added the last work about the role of pain in diagnostic delay [ref.110]

Finalyy, we have improved the clarity and readability of several tables. 

Round 2

Reviewer 2 Report

Comments and Suggestions for Authors

The text has been partially  reduced and is clearer and more comprehensible.

 I appreciate the   effort of the Authors  to  dive in a deep manner the topic of the diagnostic and therapeutic delay in STS. 

The Bibliography is  complete and adjourned.